# What is the Priestley-Taylor Wet-Surface Evaporation Parameter? Testing Four Hypotheses

Richard D. Crago[1], Jozsef Szilagyi[2,3], Russell J. Qualls[4]

[1]Department of Civil and Environmental Engineering, Bucknell University, Lewisburg, PA 17837, USA
[2]Department of Hydraulic and Water Resources Engineering, Budapest University of Technology and Economics, Budapest, Hungary
[3]Conservation and Survey Division, School of Natural Resources, University of Nebraska-Lincoln, Lincoln, NE, USA
[4]Department of Biological Engineering, University of Idaho, Moscow, ID USA

*Correspondence to*: Richard D. Crago (rcrago@bucknell.edu)

**Abstract.** This study compares four different hypotheses regarding the nature of the Priestley-Taylor parameter α. They are:
1) α is a universal constant; 2) the Bowen ratio ($H/LE$, where $H$ is the sensible and $LE$ is the latent heat flux) for equilibrium
(i.e. saturated air column near the surface) evaporation is a constant times the Bowen ratio at minimal advection (Andreas et al., 2013); 3) minimal advection over a wet surface corresponds to a particular relative humidity value, and 4) α is a constant fraction of the difference from the minimum value of one to the maximum value of α proposed by Priestley and Taylor (1972). Formulas for α are developed for the last three hypotheses. Weather, radiation and surface energy flux data from 171 FLUXNET eddy covariance stations were used. The condition $LE_{ref}/LE_p > 0.90$, was taken as the criterion for nearly-saturated
conditions (where $LE_{ref}$ is the reference and $LE_p$ is the apparent potential evaporation rate from the Penman (1948) equation). Daily and monthly average data from the sites were obtained. All formulations for α include one model parameter which is optimized such that the root mean square error of the target variable was minimized. For each model, separate optimizations were done for predictions of the target variables α, wet surface evaporation (α multiplied by equilibrium evaporation rate) and actual evaporation (the latter using a highly-successful version of the complementary relationship of evaporation). Overall, the
second and fourth hypotheses received the best support from the data.

## 1 Introduction

On a globe dominated by ocean surfaces, wet surface evaporation has obvious global importance (e.g., Brutsaert, 2023 p. 142; Andreas, 2013; McMahon et al. 2013; Szilagyi et al., 2014; Yang and Roderick, 2019; Tu et al., 2022). But estimates of wet surface evaporation can be valuable over land surfaces as well. For example, the GLEAM evaporation product (Miralles et al.,
2011, Martens et al., 2017) uses the Priestley and Taylor (1972) wet surface evaporation equation as the starting point for land surface evaporation. While climatic influences on wet surface evaporation rates can differ from those on transpiration (e.g., Schymanski and Or, 2015), in the GLEAM model, adjustments for water stress are made using a "multiplicative stress factor"

(Martens et al., 2017). Many other models and data products use some form of the Penman (1948) or the related Penman-Monteith equation (Monteith, 1965; Allen et al., 1998; McMahon et al., 2013). The advection-aridity version (Brutsaert and Stricker, 1979) of Bouchet's (1963) Complementary Relationship (CR) between actual and apparent potential evaporation (Brutsaert, 2005, p. 136) makes use of both the Priestley-Taylor and the Penman equation to estimate actual land-surface evaporation from non-saturated surfaces.

Both the Penman (1948) and the Priestley-Taylor (1972) equation are estimates of potential evaporation, the hypothetical evaporation rate that one would get from a land surface if the surface was saturated (Brutsaert; 2005, p. 136). The Penman equation consists of a radiative term and an advective term. Slatyer and McIlroy (1961, p. 3-73) noted that the advective term would be zero if the surface and the lower atmosphere were fully saturated (see further discussion in section 2.1). Evaporation under this condition is known as equilibrium evaporation. The Priestley-Taylor equation multiplies the equilibrium evaporation rate by a factor α (where α >1) to account for the presence of some vapor pressure deficit even under conditions of minimal advection--what Priestley and Taylor (1972) termed "the absence of advection".

Some work regarding α for wet surfaces treated it as a global constant to be found through field experiments (e.g., Priestley and Taylor, 1972). Other work made use of mixed-layer models of the atmospheric boundary layer (ABL), linked to a surface layer model in order to assess the role of ABL development on the value of α (de Bruin, 1983; McNaughton, 1976; McNaughton and Spriggs, 1989; Lhomme, 1997a,b, c.f., Raupach 2000). As a whole, this work suggests that ABL processes result in variability of the value of α. Since Priestley and Taylor (1972) found a single central value for α, such variability could cast doubt on the concept of a minimal-advection, wet-surface evaporation rate that can be reliably estimated. Eichinger et al. (1996) derived an explicit equation for α. Szilagyi et al. (2014) used global ocean data products to investigate α over the oceans of the world. They found a discernible relationship between α and temperature (see also Yang and Roderick, 2019). Han et al. (2021) began with the sigmoid generalized complementary equation developed by Han and Tian (2018) and modified it to apply to wet surface evaporation.

The context of this work is the estimation of wet-surface evaporation, whether under actually-wet conditions or the hypothetical evaporation rate if an unsaturated surface was actually saturated (Thornthwaite, 1948). Such hypothetical wet-surface evaporation estimates are commonly used, particularly in models based on the CR (e.g., Brutsaert and Stricker, 1979; Szilagyi and Joszsa, 2008; Crago et al. 2016; Han and Tian, 2018, 2020). In this context, it is not immediately apparent whether formulae derived for wet surfaces will also provide good results (for hypothetical wet-surface evaporation) from unsaturated surfaces. A different context is sometimes seen in the literature, in which α is essentially a moisture-availability factor (e.g., De Bruin, 1983). This latter use of α will not be further considered here.

The objective here is to gain a better conceptual understanding of α (c.f., Crago and Qualls, 2013). Similar to the limitations applied by Priestley and Taylor (1972; c.f. Andreas et al., 2013), only cases where both sensible and latent heat fluxes are positive will be considered.

Four hypotheses regarding α will be examined:

Hypothesis 1: The ratio (α) between wet-surface evaporation under minimally-advective conditions and under equilibrium conditions (i.e., a saturated atmospheric column near the wet surface) is a global constant.

Hypothesis 2: There is a globally-constant ratio (Andreas et al., 2013; Yang and Roderick, 2019) between: 1) The Bowen ratio that occurs under minimal advection with a saturated surface and 2) the Bowen ratio that would occur under equilibrium evaporation conditions, and α can be derived using this constant ratio.

Hypothesis 3: There is a globally-constant relative humidity value that can be used to derive an estimate of α that corresponds
to minimally-advective conditions

Hypothesis 4: The parameter α is a globally-constant fraction of the gap between the minimum value of one and the maximum-allowable value of α proposed by Priestley and Taylor (1972).

All four hypotheses will be examined under actually-saturated conditions, but they will also be evaluated under unsaturated conditions. Because there is no measured or reference value for the hypothetical wet-surface evaporation rate, instead the hypothetical rate will be included in a well-tested CR model for actual evaporation. That is, the CR model accuracy will be taken as an indirect measure of the method's ability to estimate the hypothetical wet-surface evaporation rate.

## 2 Theoretical Background

### 2.1 Wet surface evaporation equations

Latent heat flux $LE$ (W m$^{-2}$) is related to evaporation rate E (kg m$^{-2}$ s$^{-1}$) as $LE=l_vE$, where $l_v$ is the latent heat of evaporation (J kg$^{-1}$). Penman's (1948) equation for apparent potential evaporation ($LE_p$) from a wet surface can be written:

$$LE_p = \frac{\Delta}{\Delta + \gamma}(R_n - G) + l_v\frac{\gamma}{\Delta + \gamma}E_A , \qquad (1)$$

where $\Delta=de^*/dT$ (Pa K$^{-1}$) is evaluated at the air temperature $T_a$ (K) at height $z_T$ (m), $e$ (Pa) is vapor pressure, and $e^*$ (Pa) is
saturated vapor pressure, where both $e$ and $e^*$ are calculated using the formulations given by Andreas et al. (2013) which are valid for temperatures both above and below freezing. The net radiation is $R_n$ (W m$^{-2}$), $G$ (W m$^{-2}$) is the ground heat flux, the

latent heat of evaporation, $l_v$, is also calculated with a formulation given by Andreas (2013), $\gamma = c_p p / (0.622 l_v)$ (Pa K$^{-1}$) is the psychrometric constant, $p$ (Pa) is atmospheric pressure, and $c_p$ is specific heat of air at constant pressure (J kg$^{-1}$ K$^{-1}$). The formulae adapted from Andreas et al. (2013) have been included in the Supplement. The drying power of the air $E_A$ (kg m$^{-2}$ s$^{-1}$) is defined by:

$$E_A = f(u)[e^*(T_a) - e_a].\qquad(2)$$

Where $e_a$ is the vapor pressure at height $z_T$ and $f(u)$ (s/m) is a function of wind speed. The wind function can be calculated (Brutsaert, 2015) using Monin-Obukhov Similarity theory (MOS theory):

$$f(u) = \frac{0.622\, k^2 u}{R_d T_a ln[(z_T - d_0)/z_{0v}] ln[(z_u - d_0)/z_0]},\qquad(3)$$

where k=0.4 (dimensionless) is von Karman's constant, $R_d$ (J kg$^{-1}$ K$^{-1}$) is the ideal gas constant of dry air, $u$ (m s$^{-1}$) is wind speed measured at height $z_T$, $d_0$ (m) is the displacement height, and $z_0$ (m) and $z_{0v}$ (m) are the roughness lengths for momentum and sensible heat, respectively. Equation (3) is based on MOS theory, the standard formulation of flux-gradient relationships in the lower atmosphere (Stull, 1988, p. 376, Brutsaert, 2005, p. 128), but Penman (1948) recommended the form $f(u)=c_1(1+c_2 u_2)$ where $c_1$ and $c_2$ are empirical constants and $u_2$ (m s$^{-1}$) is wind speed at 2 m (m s$^{-1}$). This latter formulation (not used here) is preferred by some authors (e.g., Szilagyi et al., 2019) because information about the roughness of the surface (needed for $z_0$, $z_{0v}$ and $d_0$) is not needed.

Note that other versions of $LE_p$ are available, including one (e.g., Qualls and Crago, 2020; Crago and Qualls, 2021) which is based on the surface energy budget with mass and energy transport functions for the latent and sensible heat fluxes, respectively. While (1)-(3) are based on the same principles, Penman's (1948) derivation involved his well-known approximation that $\Delta$ for a wet surface is approximately equal to the ratio of the difference in vapor pressure between the surface and measurement height to the difference in temperature between the same two levels, which allowed the simple two-term equation (1). Only (1) will be used for $LE_p$ in this project.

As described by Brutsaert (2005, p. 129), air in the lowest layers blowing for a long distance over a wet surface would likely become increasingly humid. If it should approach saturation, the second term of (1) would go to zero, leaving the first term of (1) as an effective "lower limit" or "equilibrium value" for wet-surface evaporation (Slatyer and McIlroy, 1961, p. 3-73). This evaporation rate is often termed equilibrium evaporation (e.g., Brutsaert, 2005, p. 129). Here, this lower limit $LE_e$ (W m$^{-2}$) is calculated as:

$$LE_e = \frac{\Delta_{T0}}{\Delta_{T0} + \gamma}(R_n - G).\qquad(4)$$

where $\Delta_{T0}$ is $\Delta$ evaluated at the wet surface temperature $T_0$ (to be defined shortly). While $\Delta$ is commonly estimated at $T_a$, (e.g., Brutsaert, 2005, p. 126), (4) corresponds to the definition of equilibrium evaporation suggested by Andreas et al. (2014) and Qualls and Crago (2020). Namely, it is the lowest wet-surface evaporation rate possible for a given available energy value ($R_n$-

$G$) with a surface temperature of $T_0$ (K). It is a minimum because lower evaporation rates would require the vapor pressure to

exceed the saturation value (e.g., Philip, 1987; Andreas et al., 2013; Qualls and Crago, 2020). The fact that super-saturation cannot occur during evaporation explains why wet surface evaporation is limited by (4) rather than simply by ($R_n$-$G$) (see Qualls and Crago, 2020). The Bowen ratio ($Bo=H/LE;$ dimensionless) corresponding to (4) is $Bo^* = (R_n\text{-}G\text{-}LE_e)/LE_e$.

Priestley and Taylor (1972) introduced the parameter $\alpha=LE/LE_e$ (dimensionless) so that:

$$LE_{PT} = \alpha LE_e \ , \tag{5}$$

Where $LE_{PT}$ (W m$^{-2}$) estimates minimum-advection wet-surface latent heat flux. Because of some dry advection even over extensive saturated surfaces, they found $\alpha>1$. Their data suggested a typical value of $\alpha\approx1.26$. Because $LE_{PT}\geq LE_e$ and $H\geq0$, the limits on $\alpha$ are (Priestley and Taylor, 1972):

$$1 \ \leq \ \alpha \ \leq \ 1+\frac{\gamma}{\Delta_{T0}} \tag{6}$$

Hypothesis 1 suggests that (5), with $\alpha$ a global constant, defines minimal-advection wet-surface evaporation.

Andreas et al. (2013) examined thousands of measurements taken over extensive water and ice surfaces for which $H>0$ and $LE>0$ and suggested that $Bo$ is related to $Bo^*$ by:

$$Bo = a_A Bo^* , \tag{7}$$

where $a_A$ (dimensionless) was found to be a global constant of about 0.4. This is equivalent to a Priestley-Taylor $\alpha$ of:

$$\alpha_A = \frac{\Delta_{T0} + \gamma}{\Delta_{T0} + a_A\gamma} \quad . \tag{8}$$

In (8), $a_A$ is a constant, and $\Delta$ is a function of the skin temperature $T_0$ so that a discernible relationship between $\alpha$ and $T_0$ is implied by (8) (c.f., Szilagyi et al., 2014). Hypothesis 2 suggests that (8) captures the foundational concept of $\alpha$.

Yang and Roderick (2019) made a similar proposal to (7), resulting in $a_A$=0.24 based on global ocean data products. However, they noted that, in practice, $LE$ and $R_n$ cannot be known independently of each other over oceans, since increased $LE$ reduces the ocean surface skin temperature, which reduces outgoing longwave radiant fluxes, thereby increasing $R_n$. Their value of $a_A$ accounts for adjustments in the available energy resulting from this linkage. The present study assumes that $R_n$-G is known via measurements at each site.


Eichinger et al. (1996) had already proposed a dimensionless variable $C=[e^*(T_a)\text{-}e_a]/[e^*(T_0)\text{-}e_a]$ for use in an explicit method (their equation 7) to estimate $\alpha$ for wet surfaces. Plans to include an additional hypothesis based on their equation (7) in this study were abandoned when it became apparent that their $C$ (taken as a constant model parameter rather than calculated with the definition given in the previous sentence) is mathematically-equivalent to (1-$a_A$). While we will refer to (8) as the Andreas

et al (2013) formula, we acknowledge the prescient contribution of Eichinger et al. (1996).

As an alternative to (8), if there is minimum advection over a wet surface, both (1) and (5) should give the correct evaporation rate. By setting them equal to each other, one arrives at:

$$\alpha_{RH} = 1 + \left(\frac{\gamma}{\Delta_{T0}}\right)\frac{l_v f(u) e^*(T_0)(1 - RH)}{R_n - G} \ . \qquad (9)$$

where $RH$ (dimensionless) is the relative humidity of the air and $\alpha_{RH}$ is dimensionless. The values of $l_v$, $\Delta_{T0}$ and $e^*$ could all be evaluated at the wet surface skin temperature $T_0$. Equation (9) gives the correct value of $\alpha$ within the accuracy of Penman's (1948) assumption regarding $\Delta$, provided $RH$ is the measured relative humidity. However (9) is proposed here as a parameterization of $\alpha$ for both actually- and hypothetically-saturated surfaces, where $RH$ is the model parameter representing the relative humidity under saturated surface and minimal advection conditions. Small values of $(R_n\text{-}G)$ could result in unreasonably-large values of $\alpha$. Therefore, the limits given by (6) are applied to estimates of $\alpha_{RH}$. That is, if $\alpha_{RH} > 1+\gamma/\Delta$, it is set to $1+\gamma/\Delta$, and if $\alpha_{RH} < 1$, then it is set to 1.

The limits on $\alpha$ given by (6) suggest that perhaps $\alpha$ takes a constant intermediate position in between the limits. Thus, the parameter $m$ (dimensionless) is:

$$m = \frac{\alpha - 1}{\left(1 + \frac{\gamma}{\Delta_{T0}}\right) - 1} \qquad (10)$$

Or,

$$\alpha = 1 + m\frac{\gamma}{\Delta_{T0}} \ . \qquad (11)$$

Hypothesis 4 suggests that (11) is the best explanation of $\alpha$.

Szilagyi and Jozsa (2008; see also Szilagyi and Schepers, 2014; Szilagyi et al., 2017) suggested $T_0$ could be found by setting two expressions for the Bowen ratio (here, given by $H/LE_p$) equal to each other:

$$\frac{R_n - G - LE_p}{LE_p} = \gamma\frac{T_0 - T_a}{e^*(T_0) - e_a} \qquad (12)$$

where the equation used for $e^*(T_0)$ (from Andreas et al., 2014) is given in the Supplement. The wet surface temperature in (12) is $T_0$, which can be easily found from (12) with a numerical root finder. Equation (12), thus solved provides the wet surface temperature $T_0$ from data taken from either saturated or unsaturated surfaces (Szilagyi and Schepers, 2014).

## 2.2 The Complementary relationship (CR) of Evaporation

In the Complementary Relationship (CR) between actual and potential evaporation (Bouchet, 1963), regional evaporation from a saturated surface, the apparent-potential evaporation rate, and the actual evaporation rate are all identical (Brutsaert, 2015,

p. 136). According to the advection-aridity approach (Brutsaert and Stricker, 1979), apparent potential evaporation corresponds to the Penman equation (1) or to the evaporation from a small wet patch, and the wet regional surface rate corresponds to the Priestley and Taylor (1972) equation (5). As the surface dries, less water is available to evaporate, so actual evaporation decreases. This results in a drier and warmer lower atmosphere, which increases apparent potential (wet patch) evaporation. Conversely, if the lower atmosphere becomes dry and warm (in the absence of significant dry advection), this implies that regional evaporation rates are low. Thus, evaporation and apparent potential evaporation change in opposite directions—they complement each other. An estimate of the Priestley-Taylor $\alpha$ is an integral part of most CR models, and the performance of CR models making use of the four different hypotheses regarding $\alpha$ can serve as a further test of the hypotheses. Note that Han et al. (2021) took a different approach, by adapting the CR model of Han and Tian (2018) to estimate evaporation from wet surfaces; this results in a non-linear dependence of wet surface evaporation on equilibrium evaporation.

As formulated by Brutsaert (2015, p. 136), the CR can be formulated in terms of $x=LE_w/LE_p$ and $y=LE/LE_p$, where $LE_w$ is given by (5) and $LE_p$ by (1)-(3). Both $x$ and $y$ are dimensionless. Values of $\alpha$ to be used in (5) will be discussed in Section 3. Brutsaert (2015) used physical reasoning to suggest that at $x=0$, the boundary conditions are $y=0$ and $dy/dx=0$, while at $x=1$, they are $y=1$ and $dy/dx=1$. Crago et al. (2016), however, noted that $y$ can approach zero when there is no water available to evaporate, but $x$ cannot approach zero unless $LE_w$ goes to zero. The smallest $x$ can get is:

$$x_{min} = \frac{LE_w}{LE_{pmax}}, \qquad (13)$$

where $LE_{pmax}$ (W m$^{-2}$) is given by

$$LE_{pmax} = \frac{\Delta_d}{\Delta_d + \gamma}(R_n - G) + l_v \frac{\gamma}{\Delta_d + \gamma} f(u) e^*(T_d). \qquad (14)$$

In (15), the subscript "$d$" means the variable is evaluated at $T_d$, the "dry air temperature". A straight line with slope $de/dT_g=-\gamma$ (where $T_g$ is a generic temperature variable) represents an isenthalp (line of constant available energy) through ($T_a$, $e_a$) on a graph of temperature ($x$-axis) versus vapor pressure ($e$ on the y-axis). The temperature at which this isenthalp reaches $e=0$ is $T_d$ (Szilagyi et al., 2017; Crago and Qualls, 2021). That is,

$$T_d = T_a + \frac{e_a}{\gamma} . \qquad (15)$$

Crago et al. (2016) suggested $x$ could be "rescaled" through the transform:

$$X = \frac{x - x_{min}}{1 - x_{min}} \qquad (16)$$

A simple formulation suggested by Crago et al. (2016) is:

$$y = X. \qquad (17)$$

Crago et al. (2022) considered data from seven FLUXNET sites in Australia as well as global, gridded ERA5 data (Hersbach, 2020) produced by ECMWF (European Centre for Medium-Range Weather Forecasts; https://www.ecmwf.int/). With the

FLUXNET data, (17) consistently performed best at predicting reference (eddy covariance) latent heat fluxes. Since FLUXNET data are used here as well, (17) will be the assumed CR formula for this study. From (17) latent heat flux estimates can be found with $LE_{est}=X(LE_p)$

Equations (1) through (5) and CR methods are generally considered applicable at time scales ranging from daily to monthly, with monthly being most common (McMahon et al., 2013). Equations (1) through (5) require homogeneous surfaces corresponding to the spatial extent of the flux footprint (e.g., Schuepp, 1990), typically corresponding to several hundred meters, while CR formulations are best suited for homogeneous conditions at the "regional" scale (Brutsaert, 2005, p. 136) of perhaps tens of km.


## 3 Methodology

### 3.1 Data sources and processing

Monthly- and daily-average data pre-processed by FLUXNET were downloaded as csv files from the fluxnet.org website for 171 eddy covariance stations (listed in Table S1 in the Supplement). At least minimally-adequate fetches are assumed at all
sites included in FLUXNET. Measurement heights, latitudes, longitudes, IGBP land classes (www.igbp.net), and canopy heights were provided for these sites by Wang et al. (2020; see their supporting information). Wang et al. (2020) assumed $z_0=0.123h_c$, $d_0=0.67h_c$, and $z_{0v}=0.1z_0$, where $h_c$ (m) is the reported canopy height. These values were all adopted herein. Separate wind speed, temperature and humidity measurement heights were not included by Wang et al. (2020), so it is assumed here that they are all measured at the single given height. Net radiation, ground heat flux, sensible and latent heat fluxes, air
pressure, air temperature, vapor pressure deficit, and wind speed were included in the FLUXNET downloads. All variables employed some gap filling using the MDS [Marginal Distribution Sampling (Reichstein, 2005)] method as described by Pastorello et al. (2020). Data flagging, quality assurance and control for all the variables also followed the procedures outlined by Pastorello et al. (2020).

Following the procedures outlined by Pastorello et al. (2020), the half-hourly or hourly energy fluxes are also gap-filled using the MDS (Reichstein, 2005) method, and these are used by FLUXNET to derive the daily or monthly reference values used here. The FLUXNET dataset includes the variables "H_CORR" and "LE_CORR", which indicate corrected values, that is, values that correspond to energy budget closure. However, the corresponding uncorrected variables ("LE_F_MDS" and "H_F_MDS") are available for more sites and times. These latter surface fluxes were used for this study for the reference
values $LE_{ref}$ and $H_{ref}$, respectively. Issues regarding energy budget closure with eddy covariance fluxes are complicated, as discussed by Mauder et al. (2020). In this study, the downloaded values of sensible heat flux are taken to be the final reference

values $H_{ref}$, while downloaded latent heat flux values are adjusted so that monthly (daily) energy budget closure is obtained: $LE_{ref}=(R_n-G)-H_{ref}$. This was the procedure recommended by Wang et al. (2020; see also Tu et al., 2022, 2023).

Months (or days) with Eddy covariance values of $H$ and $LE$ less than zero or $R_n-G<0$ were screened out of the dataset; this eliminates periods of strong dry advection that result in negative $H_{ref}$. The ground heat flux $G$ was not measured at all for some of the sites, and missing values of G also occurred. When measurements of G were not available, a value of zero was assumed. Over a 24-hour period, G "is often near zero" (Stull, 1988), so this assumption is not unreasonable. Over the daily to monthly time-scales at which (1) and (5) are commonly used (McMahon, 2014) the assumption likely improves as the averaging time

increases.

The CR has been used at time scales from hourly to yearly (Brutsaert, 2023, p. 147), but the CR, the Penman equation (1), and the Priestley-Taylor equation (5) most typically use daily- to monthly-average values (McMahon, 2013). It is true that use of time-averages of variables as inputs to non-linear equations can lead to "significant errors" (Slatyer and McIlroy, 1961, p. 3-

58). However, CR and the Priestley-Taylor wet-surface equation both assume that that the land-surface conditions and the temperature and humidity in the lower atmosphere are well-adjusted to each other (Brutsaert, 2023, p. 147). The diurnal cycle makes this adjustment unlikely over periods less than 24 hours (McMahon, 2013). Therefore, the approach here is to use daily (monthly) average input values to produce daily (monthly) energy fluxes (e.g., Penman, 1948; McMahon, 2013; Brutsaert, 2023). That is, daily to monthly time scales are suited to these equations, as spatial scales corresponding to small watersheds

are suited to saturation-excess runoff (e.g., Chow et al., 1988).

With this dataset, the monthly (daily) mean of reference latent heat flux is 61 W m$^{-2}$ (62 W m$^{-2}$), the median is 58 W m$^{-2}$ (56 W m$^{-2}$) and the standard deviation is 41 W m$^{-2}$ (48 W m$^{-2}$). Thus, the central tendencies for monthly and daily values are similar, but the daily has more spread about the mean.


The decisions described above regarding data inclusion, screening, and correction reflect a desire to obtain a broad range of climates, land covers, and seasons, so as to test the four hypotheses under as wide a range of conditions as possible. While these decisions do entail some risk of including lower quality data, we think they are defendable as outlined above. However, much of the analysis was repeated after removing data for which the FLUXNET quality control index (which ranges from 0

for very poor to 1 for excellent) was less than 0.9, with little difference in numerical results and no difference in qualitative results (such as which methods performed better than other methods) compared to the results without filtering for data quality.

Eleven different IGBP land surface classes (e.g., Loveland et al., 1999) are in the original data. They include: Wooded Savannas (WSA); Grasslands (GRA); Evergreen Broadleaf Forests (EBF); Crops (CRO); Evergreen Needleleaf Forests (ENF);

Savannas (SAV); Deciduous Broadleaf Forests, (DBF); Closed Shrublands (CSH); Mixed Forests (MF); Open Shrublands (OSH); and Permanent Wetlands (WET). Classes for each site were provided by Wang et al. (2020).

## 3.2 Estimates of wet surface α

Determination of wet-surface values of α requires determination of data representing saturated surface conditions. Saturated or nearly-saturated land surface conditions were assumed when $LE_{ref} > T(LE_p)$ where $T$ (dimensionless) is a threshold value 290 which had to be determined. A value of $T$ was sought such that the linear regression between $LE_{ref}$ and $LE_p$ (for data for which the condition above is met) falls nearly on the one-to-one line, and at the same time root mean square (RMS) errors between $LE_{ref}$ and $LE_p$ are very small. After filtering for wet-surface conditions, $\alpha_{ref}$ was calculated as $LE/LE_e$.

Trial values of the parameters $\alpha_c$, $a_A$, $RH$, and $m$, where $\alpha_c$ is a constant (global) value of α were selected randomly from a 295 range of reasonable values (that is, α ranged from 1 to 1.6 and $a_A$, $RH$, and $m$ all varied from 0 to 1). These values were used at all sites and times satisfying the wet surface condition. Three thousand values drawn randomly from these ranges were evaluated to determine the optimal parameter values. These optimal parameter values were used to estimate different versions of $\alpha_{est}$, namely $\alpha_c$, $a_A$ from (8); $\alpha_{RH}$ from (9); and $\alpha_m$ from (11). These $\alpha_{est}$ values were compared to $\alpha_{ref}=LE_{ref}/LE_e$. The trial parameter values that minimized the root mean square difference (RMSD) between $\alpha_{est}$ and $\alpha_{ref}$ were taken to the be tuned 300 parameter values; the values of $\alpha_{est}$ will be called the best-fit values.

## 3.3 Estimates of wet surface evaporation

Randomly selected parameter values were chosen 3000 times (over the same range as in section 3.2) to estimate $\alpha_c$, $\alpha_A$, $\alpha_{RH}$, and $\alpha_m$ for use in (5) to estimate $LE_{est}=\alpha_{est}LE_e$, where $\alpha_{est}$ is estimated from the parameter values using (8), (9), and (11). This was done for all wet-surface measurements. Tuned parameter values were those giving minimum RMSD (root mean square 305 difference) between the resulting $LE_{est}$ and $LE_{ref}$ and the resulting $LE_{est}$ values are considered best-fit values. The same tuned parameter values were used for all stations and all times. Note that the tuned parameter values found in this analysis may differ from those found in section 3.2.

## 3.4 Complementary relationship for actual evaporation

Next, actual evaporation was estimated with the CR using all available data. That is, the analysis was not limited to only wet 310 surface data. Specifically, 3000 new samples of the parameter values were chosen from the same range as above. Those parameter values were substituted into (8), (9) or (11), and were used to calculate estimates of α. Those values in turn were used in (1) through (5) along with (14) through (16) in (17), and finally in $LE_{est}=y*LE_p$. The RMSD between $LE_{est}$ and $LE_{ref}$ was found for each of the four methods of estimating $LE_{PT}$. The tuned parameter values were those minimizing the RMSD and the corresponding $LE_{ref}$ values are the best-fit values.

## 4 Results

For monthly averaging times, the wet surface threshold was established to be $T$=0.90, which resulted in a regression equation with slope of about 1 and intercept very near 0, while RMS errors were small (Figure 1, panel 4). The process by which this value of T was established is described using Figure 1, and the corresponding statistics for both monthly and daily data are found in Table 1. The second row of text in each panel identifies the range of $LE_{ref}/LE_p$ values incorporated into the graph. If the lower limit of accepted values was $T$, then in the left column (panels 1, 3, and 5) an upper limit of 2-$T$ was imposed. In the right column (panels 2, 4, and 6) no upper limit was imposed (as indicated by an upper limit denoted '-'). There seems to be no compelling reason to impose an upper limit on wet-surface $LE_{ref}/LE_p$, even though the upper limit improved many of the statistics (i.e., comparing panel 1 to 2; 3 to 4, and 5 to 6). In the right column, panel 2 has a slope somewhat below 1 and panel 6 has a large RMS difference. Panel 4, where all points with $LE_{ref}/LE_p$>0.90 were included, was taken as a reasonable compromise. As shown in panel 4, wet surface evaporation defined in this way occurred on 430 months and from 50 of the sites. The sites included IGBP classes CRO, ENF, GRA, DBF, WET, OSH, and EBF. For daily averaging times, a similar process was followed (statistics included in Table 1). Wet surface evaporation (defined again by T=0.90) occurred on 22998 days, involving 158 of the 171 stations and included IGBP classes WSA, GRA, EBF, CRO, ENF, SAV, DBF, CSH, OSH, WET and MF. Figure 2 shows the location of sites having at least one month of wet surface measurements (top panel) and those having none (bottom panel).

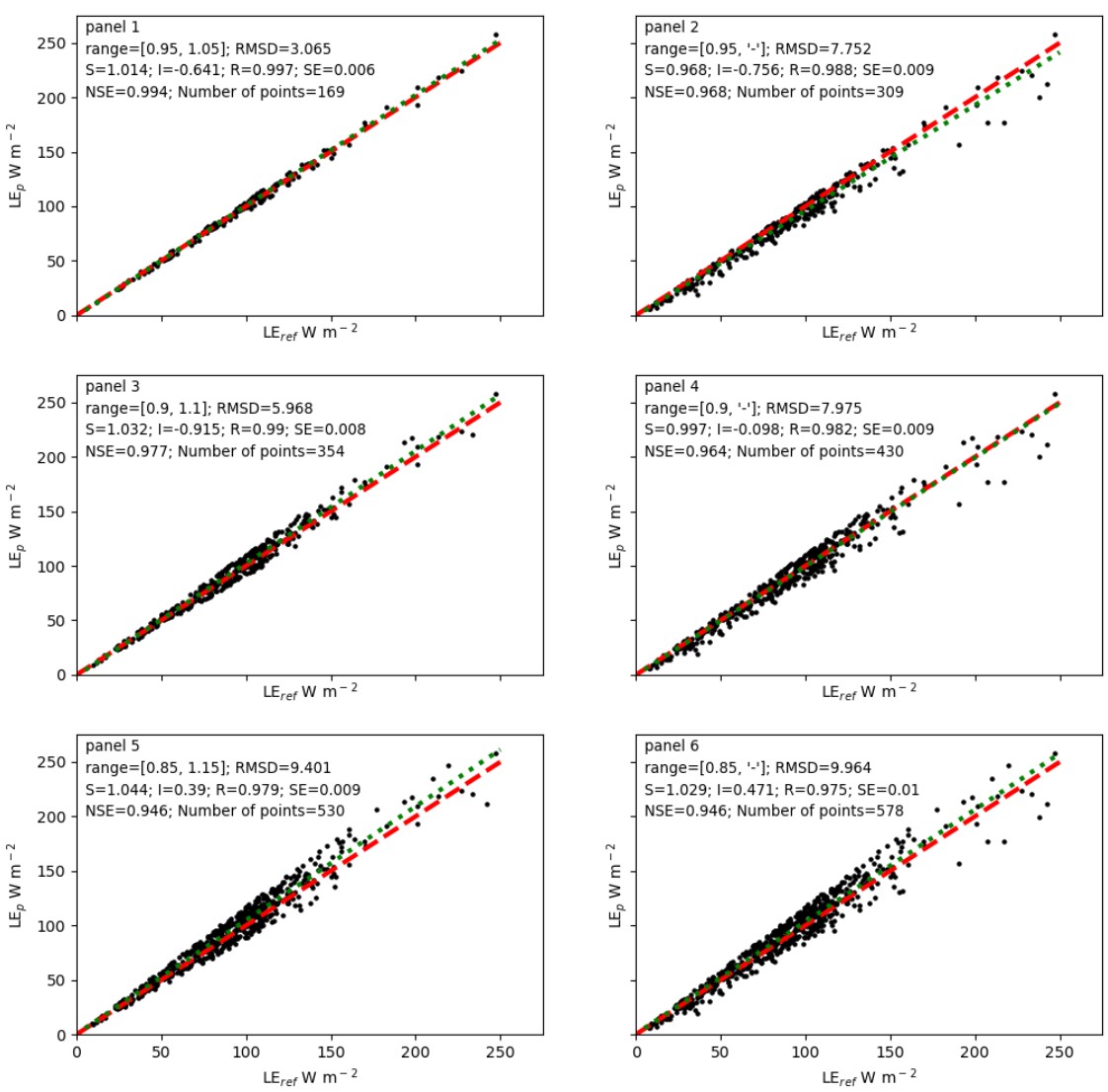

Figure 1. Comparison of reference values (LE_ref) to estimates $LE_{est}$ from equation (1) for various threshold (*T*) values to define wet surface conditions for monthly evaporation. In each plot, only data from a given

range of $LE_{ref}/LE_p$ are plotted and included in the statistics included in the upper left corner of each plot. The left column of plots applies both upper and lower limits on $LE_{ref}/LE_p$, while the right column applies only a lower limit. Panel 4 was taken as the best compromise, as it has many desired features, including a relatively large numbers of points, regression slopes and intercepts near zero, and low rms differences. A total (panel 4) of 430 months of wet surface evaporation were identified. The red line is one-to-one and the blue line is the linear regression. The notation RMSD is root mean square difference, R is correlation coefficient, NSE is Nash-Sutcliffe efficiency (Nash and Sutcliffe, 1970), and S and I are the slope and intercept in the linear regression equation. The "range" in the second line of text in each plot indicates the range of $LE_p/LE_{ref}$ values included in the analysis, with the first number in brackets indicating the lower and the second number the higher limit of the range; the notation '-' indicates no upper limit. The red dashed line is one-to-one, and the green dotted line represents the linear regression.

Table 1. Results of regression between $LE_p$ and $LE_{ref}$ for various ranges of $LE_{ref}/LE_p$

| Averaging time | Range of $LE_{ref}/LE_p$ | No. points | R [1] | Slope [1] | Intercept [1] (W m⁻²) | NSE [1] | RMSD [1] (W m⁻²) |
|---|---|---|---|---|---|---|---|
| Monthly | 0.95-1.05 | 169 | .997 | 1.014 | -0.641 | 0.994 | 3.1 |
| Monthly | 0.90-1.10 | 354 | 0.99 | 1.032 | -0.915 | 0.977 | 6.0 |
| Monthly | 0.85-1.15 | 530 | 0.979 | 1.044 | 0.39 | 0.946 | 9.4 |
| Monthly | 0.95- | 309 | 0.988 | 0.968 | -0.756 | 0.968 | 7.8 |
| Monthly | 0.90- | 430 | 0.982 | 0.997 | -0.098 | 0.964 | 8.0 |
| Monthly | 0.85- | 578 | 0.975 | 1.029 | 0.471 | 0.946 | 10.0 |
| Daily | 0.95-1.05 | 7126 | 0.998 | 1.007 | -0.202 | 0.996 | 3.0 |
| Daily | 0.90-1.10 | 14396 | 0.993 | 1.021 | -0.406 | 0.985 | 6.2 |
| Daily | 0.85-1.15 | 21210 | 0.986 | 1.044 | -0.744 | 0.986 | 9.4 |
| Daily | 0.95- | 18456 | 0.983 | 0.961 | -3.446 | 0.948 | 11.2 |
| Daily | 0.90- | 22986 | 0.981 | 1.000 | -3.648 | 0.957 | 10.8 |
| Daily | 0.85- | 27796 | 0.977 | 1.036 | -3.621 | 0.953 | 11.7 |

[1] R: correlation coefficient; Slope, Intercept: S and I in $LE_p=S*LE_{ref}+I$; NSE: Nash and Sutcliffe (1970) efficiency; RMSD: root mean square difference.

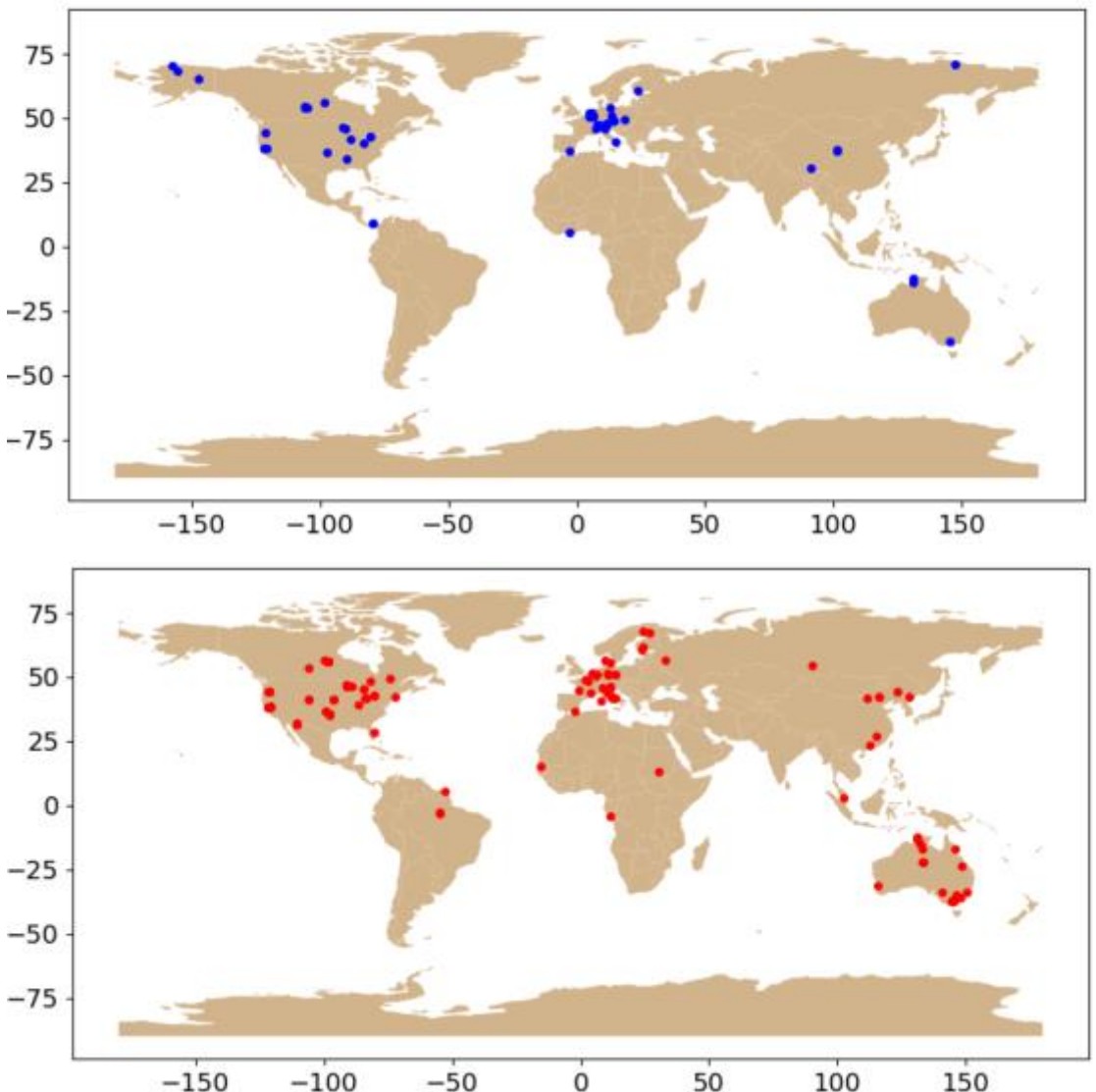

Figure 2. Global distribution of sites with some monthly measurements classified as wet surface evaporation (top panel) and those with no wet surface values (bottom panel). For daily average data, many more sites had some days of wet surface evaporation, so for daily averaging, the top panel would have more data points and the bottom panel fewer. The background map is the 'naturalearth_lowres' basemap provided by geopandas (geopandas.org).

As described in section 3.2, using only wet-surface measurements, tuned values of $\alpha_c$, $a_A$, $RH$, and $m$ were found that minimized RMSD between reference and estimated values of $\alpha$. Different tuned values of these parameters were also found for use in estimating wet surface evaporation using the Priestley-Taylor equation (5). Results for estimating $\alpha$ under wet-surface
conditions are found in panel a of Figure 3, and results for estimating wet surface evaporation itself are shown in panel b; both sets of results are also provided in Table 2. Finally, still-different tuned values of $\alpha_c$, $a_A$, $RH$, and $m$ were those which produced the minimum values of RMSD between $LE_{est}$ and $LE_{ref}$ using the CR formulation (17). That is, $LE_{est}$ is found by taking $y$ found with (17), where $LE_w$ is given by (5), $LE_p$ by (1) through (3), and $y$ by (13) through (17). This $y$ was multiplied by $LE_p$ from (1) to get $LE_{est}$. The tuned parameters that result when the goal is to obtain the best fit between $\alpha_{est}$ and $\alpha_{ref}$, are different than
those when the goal is to obtain the best fit between wet-surface $LE_{PT}$ and $LE_{ref}$, and in turn these are different than when the goal is to obtain best fit between $LE_{est}$ and $LE_{ref}$ using (17). Reasons and implications for these differences will be discussed in section 5.

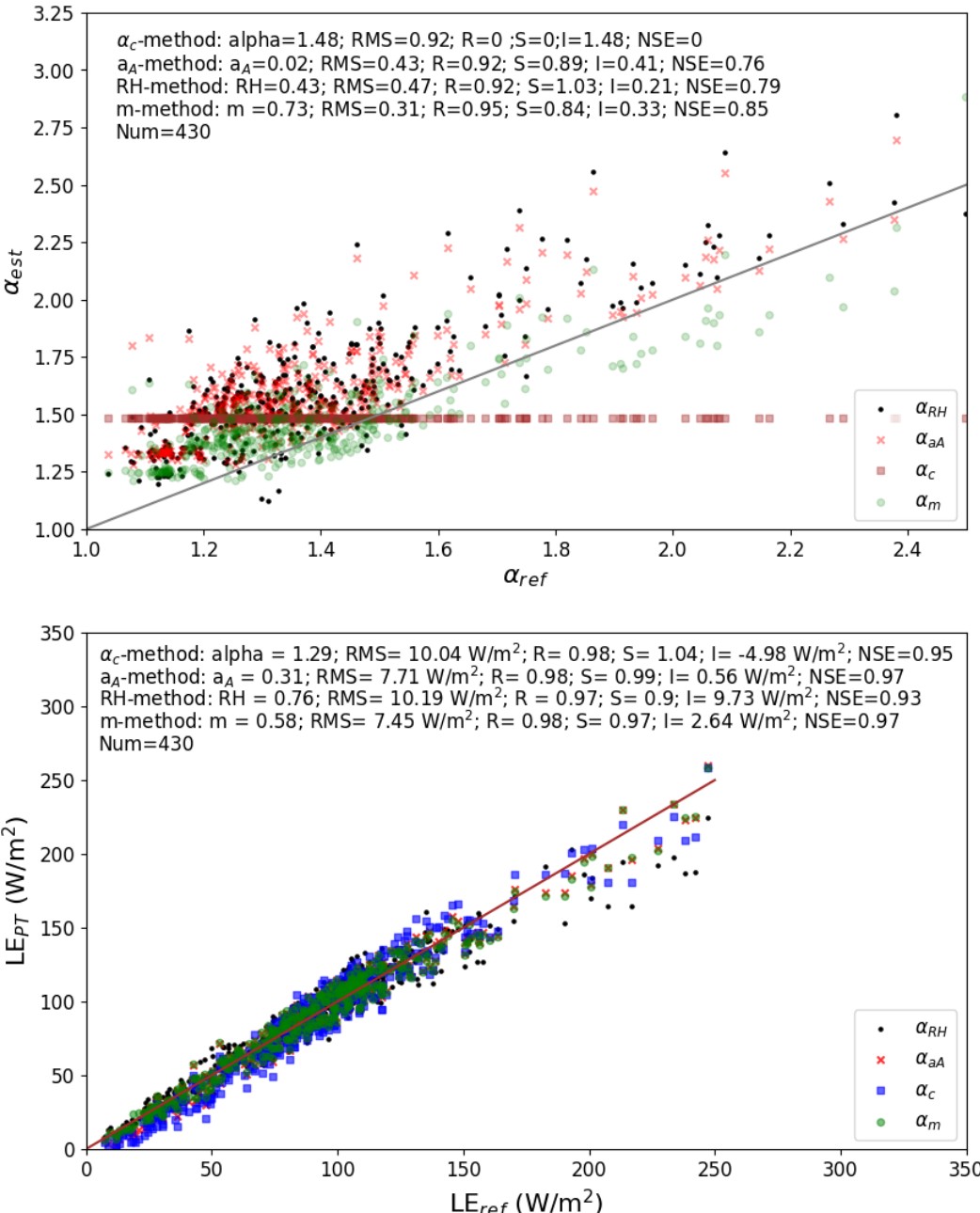

Figure 3. Results for monthly estimates of wet-surface α and of wet-surface *LE*. Parameter values and statistics are included at the top of each graph. RMSD is root mean square error, R is correlation coefficient, NSE is the Nash and Sutcliffe (1970) efficiency, and *S* and *I* are the coefficients in the linear regression equation.

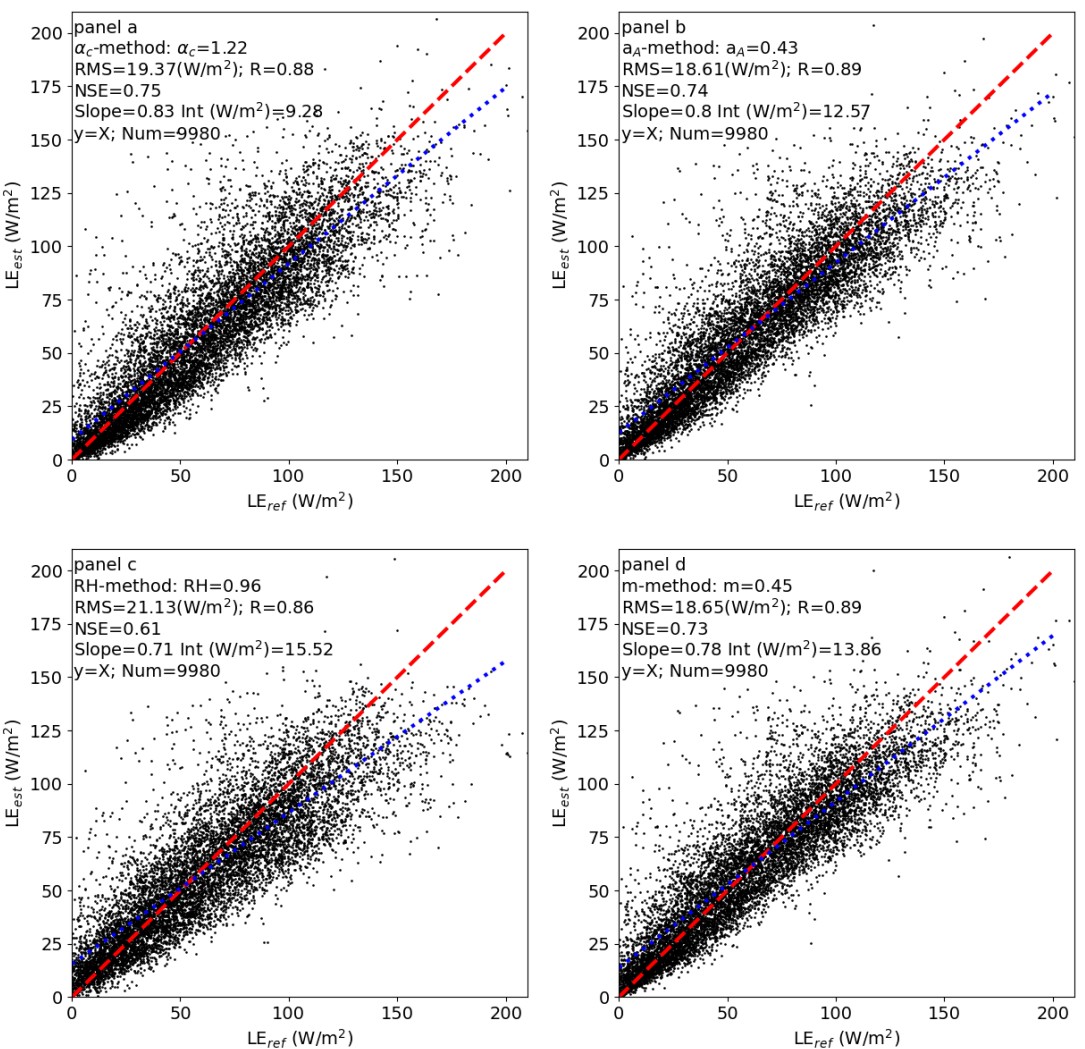

Figure 4. Results for monthly estimates of $LE_{ref}$ using the CR. Panel a uses the $\alpha_c$-method, panel b the $a_A$-method, panel c the RH-method, and panel d the m-method. Parameter values and statistics are included at the top of each graph. RMS is root mean square error, R is correlation coefficient, NSE is the Nash and Sutcliffe (1970) efficiency, and Slope and Int are the coefficients in the linear regression equation $y=Slope*x+Int$, where $x$ is $LE_{ref}$ and $y$ is $LE_{est}$. 'Num' is the number of data points included. All IGBP classes that are in the dataset are included. Red dashed line is one-to-one, blue dotted line is the linear regression.

Table 2. Summary of results for monthly data (430 wet surface months; 9980 months total)

| Method | Statistic to be minimized | Tuned variable | Parameter value | RMSD | [1] R | [1] S | [1] I | [1] NSE |
|---|---|---|---|---|---|---|---|---|
| $\alpha_c$ | RMSD([2] $\alpha_{est}$, $\alpha_{ref}$) | $\alpha_c$ | 1.48 | 0.92 | 0 | 0 | 1.48 | 0 |
| $a_A$ | RMSD([2] $\alpha_{est}$, $\alpha_{ref}$) | $a_A$ | 0.02 | 0.43 | 0.92 | 0.89 | 0.41 | 0.76 |
| $RH$ | RMSD([2] $\alpha_{est}$, $\alpha_{ref}$) | $RH$ | 0.43 | 0.47 | 0.92 | 1.03 | 0.21 | 0.79 |
| $m$ | RMSD([2] $\alpha_{est}$, $\alpha_{ref}$) | $m$ | 0.73 | 0.31 | 0.95 | 0.84 | 0.33 | 0.85 |
|  |  |  |  | W/m$^2$ |  |  | W/m$^2$ |  |
| $\alpha_c$ | RMSD([3] $LE_{w\_est}$, $LE_{ref}$) | $\alpha_c$ | 1.29 | 10.04 | 0.98 | 1.04 | -4.98 | 0.95 |
| $a_A$ | RMSD([3] $LE_{w\_est}$, $LE_{ref}$) | $a_A$ | 0.31 | 7.71 | 0.98 | 0.99 | 0.56 | 0.97 |
| $RH$ | RMSD([3] $LE_{w\_est}$, $LE_{ref}$) | $RH$ | 0.76 | 10.19 | 0.97 | 0.9 | 9.73 | 0.93 |
| $m$ | RMSD([3] $LE_{w\_est}$, $LE_{ref}$) | $m$ | 0.58 | 7.45 | 0.98 | 0.97 | 2.64 | 0.97 |
|  |  |  |  | W/m$^2$ |  |  | W/m$^2$ |  |
| $\alpha_c$ | RMSD([4] $LE_{est}$, $LE_{ref}$) | $\alpha_c$ | 1.22 | 19.37 | 0.88 | 0.83 | 9.28 | 0.75 |
| $a_A$ | RMSD([4] $LE_{est}$, $LE_{ref}$) | $a_A$ | 0.43 | 18.61 | 0.89 | 0.74 | 12.57 | 0.74 |
| $RH$ | RMSD([4] $LE_{est}$, $LE_{ref}$) | $RH$ | 0.96 | 21.13 | 0.86 | 0.71 | 15.56 | 0.61 |
| $m$ | RMSD([4] $LE_{est}$, $LE_{ref}$) | $m$ | 0.45 | 18.65 | 0.89 | 0.78 | 13.86 | 0.73 |

[1] RMSD: root mean square difference; R: correlation coefficient; Slope, Intercept: S and I in $LE_p=S*LE_{ref}+I$; NSE: Nash and Sutcliffe (1970) efficiency.

[2] RMSD($\alpha_{est}$, $\alpha_{ref}$) is the RMSD for between $\alpha$ estimates found with the prescribed method and the reference values found from $\alpha_{ref}=LE_{ref}/LE_e$ for times with wet surface evaporation conditions.

[3] RMSD($LE_{w\_est}$, $LE_{ref}$) is the RMSD for between $LE_{est}$ estimates found with the prescribed method and the reference values $LE_{ref}$ for times with wet surface evaporation conditions.

[4] RMSD($LE_{est}$, $LE_{ref}$) is the RMSD for all wetness conditions between estimates $LE_{est}$ and $LE_{ref}$.

Table 3. Summary of results for daily data (22998 wet surface days; 276020 days total)

| Method | Statistic to be minimized | Tuned variable | Parameter value | RMSD | [1] R | [1] S | [1] I | [1] NSE |
|---|---|---|---|---|---|---|---|---|
| $\alpha_c$ | RMSD([2] $\alpha_{est}$, $\alpha_{ref}$) | $\alpha_c$ | 1.58 | 1.31 | 0 | 0 | 1.58 | 0 |
| $a_A$ | RMSD([2] $\alpha_{est}$, $\alpha_{ref}$) | $a_A$ | 0.01 | 0.59 | 0.92 | 0.92 | 0.43 | 0.79 |
| *RH* | RMSD([2] $\alpha_{est}$, $\alpha_{ref}$) | *RH* | 0 | 0.93 | 0.75 | 0.65 | 0.84 | 0.34 |
| *m* | RMSD([2] $\alpha_{est}$, $\alpha_{ref}$) | *m* | 0.71 | 0.42 | 0.95 | 0.86 | 0.3 | 0.87 |
| | | | | W/m$^2$ | | | W/m$^2$ | |
| $\alpha_c$ | RMSD([3] $LE_{w\_est}$, $LE_{ref}$) | $\alpha_c$ | 1.29 | 12.65 | 0.97 | 1.03 | -5.27 | 0.94 |
| $a_A$ | RMSD([3] $LE_{w\_est}$, $LE_{ref}$) | $a_A$ | 0.31 | 9.45 | 0.98 | 1 | 0.05 | 0.97 |
| *RH* | RMSD([3] $LE_{w\_est}$, $LE_{ref}$) | *RH* | 0.74 | 12.43 | 0.97 | 0.93 | 7.96 | 0.93 |
| *m* | RMSD([3] $LE_{w\_est}$, $LE_{ref}$) | *m* | 0.57 | 9.01 | 0.98 | 0.97 | 1.95 | 0.97 |
| | | | | W/m$^2$ | | | W/m$^2$ | |
| $\alpha_c$ | RMSD([4] $LE_{est}$, $LE_{ref}$) | $\alpha_c$ | 1.18 | 25.03 | 0.86 | 0.76 | 14.15 | 0.66 |
| $a_A$ | RMSD([4] $LE_{est}$, $LE_{ref}$) | $a_A$ | 0.52 | 24.71 | 0.86 | 0.74 | 16.36 | 0.65 |
| *RH* | RMSD([4] $LE_{est}$, $LE_{ref}$) | *RH* | 0.97 | 26.67 | 0.84 | 0.66 | 18.76 | 0.51 |
| *m* | RMSD([4] $LE_{est}$, $LE_{ref}$) | *m* | 0.36 | 24.81 | 0.86 | 0.72 | 17.39 | 0.63 |

[1] RMSD: root mean square difference; R: correlation coefficient; Slope, Intercept: S and I in $LE_p=S*LE_{ref}+I$; NSE: Nash and Sutcliffe (1970) efficiency.

[2] RMSD($\alpha_{est}$, $\alpha_{ref}$) is the RMSD for between $\alpha$ estimates found with the prescribed method and the reference values found from $\alpha_{ref}=LE_{ref}/LE_e$ for times with wet surface evaporation conditions.

[3] RMSD($LE_{w\_est}$, $LE_{ref}$) is the RMSD for between $LE_{est}$ estimates found with the prescribed method and the reference values $LE_{ref}$ for times with wet surface evaporation conditions.

[4] RMSD($LE_{est}$, $LE_{ref}$) is the RMSD for all wetness conditions between estimates $LE_{est}$ and $LE_{ref}$.

## 5 Discussion

### 5.1 General trends

For convenience, the use of α with a single global value will be called the "$\alpha_c$-method" (corresponding to Hypothesis 1), with (8) it will be the "$a_A$-method" (Hypothesis 2), with (9) it will be the "$RH$-method" (Hypothesis 3), and with (11) it will be the "$m$-method" (Hypothesis 4). After discussing trends found in the results, the four hypotheses will be evaluated based on the results.

Figure 1 shows that $LE_{ref}$ and $LE_p$ (1) are very similar when the threshold for wet surfaces at a monthly time scale is taken to be $T=LE_{ref}/LE_p=0.90$. For daily data the same threshold provides good results, with low RMS error and linear regression close to the one-to-one line (Table 1), so $LE_{ref}/LE_p>0.9$ was chosen as the indicator of wet surface evaporation, with no upper threshold to $LE_{ref}/LE_p$. Figure 2 shows the geographical location of sites that had some wet-surface evaporation months (upper panel) and sites that did not (bottom panel).

Figure 3 gives the results from the four methods in terms of prediction of $\alpha_{ref}$ itself (top panel) and in wet surface evaporation estimates from (5) (bottom panel). Results from the use of the four methods when used in the CR model $y=X$ are shown in Figure 4. Even when estimates of α differ considerably from the reference values (Figure 3, top panel), the methods still provide good estimates of wet surface evaporation ($LE_w$—Figure 3 bottom panel) and actual evaporation (Figure 4).

Table 2 provides much of the same data as Figures 3 and 4 (for monthly averaging), while Table 3 provides the same information for daily averaging. A large number of FLUXNET sites spanning a wide range of climates and land cover classes were included in this study; such a diversity and large number of sites provides some confidence that the trends discussed here would apply to other sites and regions.

Different tuned parameter values were found to calculate $\alpha_{est}$ (Figure 3, top panel) $LE_{PT}$ (Figure 3, bottom panel), and $LE_{est}$ (Figure 4). Ideally, the tuned parameter values would remain nearly identical in the three cases. A likely explanation for this difference is as follows: In the top panel of Figure 3, all $\alpha_{ref}$ values count equally in determination of tuned parameter values that produce the best-fit $\alpha_{est}$ for each of the methods. But in the bottom panel of Figure 3, $\alpha_{est}$ values that correspond to small values of ($R_n$-$G$) have far less influence on the RMSD of $LE_{PT}$ than those corresponding to larger $R_n$-$G$. If $R_n$-$G=5$ W m$^{-2}$, an increase in alpha from 1.1 to 1.3 only increases $\alpha(R_n$-$G)$ from 5.5 to 6.5 W m$^{-2}$, whereas if $R_n$-$G$ is 200 W m$^{-2}$, it increases from 220 to 260 W m$^{-2}$, so that the larger $R_n$-$G$ would influence RMSD of $LE_{PT}$ more. Similarly, when moving to the CR estimate $LE_{est}$ in Figure 4, the CR estimates again apply different weight to the various estimates of α, so that different tuned parameter

values result here as well. Actually, Brutsaert (2023, p. 149) treats the parameter $\alpha$ in (5) as a completely-different parameter from the $\alpha$ embedded in (17). While the present authors consider both to be the same parameter, we recognize that its tuned value could vary depending on the context.

For reference, when the parameter values used in Figure 3 (lower panel) are used in the CR (namely $\alpha=1.29$ for the $\alpha_c$-method;
$\alpha_A=0.31$ for the $\alpha_A$-method, $RH=0.76$ for the $RH$-method, and $m=0.58$ for the $m$-method), RMS errors increased (from 19.37 to 20.12; from 18.61 to 19.25; from 21.13 to 26.84; from 18.65 to 19.29, respectively). Note that the $\alpha_A$- and $m$-methods still provide the lowest RMSD values.

Figure 3 and Tables 2 and 3 provide evidence that all four methods provide acceptable estimates of actual wet surface
evaporation rates? But what about estimation of hypothetical wet surface evaporation rates? Szilagyi and Jozsa (2008) and Szilagyi and Schepers (2014) have provided good evidence that a small wet patch within a drying region with wind speed and available energy held constant should maintain a constant surface temperature. The use of (12) to get the temperature of an actually-saturated region is straightforward. Crago and Qualls (2021), Qualls and Crago (2020) and Szilagyi (2021) show graphically, using $e$ versus $T$ graphs, how air and ground-surface isenthalps [lines of constant available energy on a $(T, e)$
graph] can be determined, and they show that $T_0$ is simply the intersection of the ground-surface isenthalp with the saturation vapor-pressure curve.

There are three distinct explanations for why a best-fit estimate $LE_{est}$ using the CR (17) might differ from the $LE_{ref}$ values: First, $\alpha$ may not be estimated correctly (see the preceding paragraphs); second, the $LE_{PT}$ estimate may not adequately represent
the hypothetical wet-surface evaporation rate; and third, the CR formulation may be inadequate. No method to distinguish the effects of these is apparent, so that the results in Figure 4 and Tables 2 and 3 provide only an indirect test of the adequacy of the four methods to estimate $\alpha$ for hypothetical wet surfaces.

Under drying surface conditions, since the wet surface temperature remains constant during drying (assuming $R_n$-$G$ and wind
speed are constant—see Szilagyi and Schepers, 2014), $T_0$ found with (12) under drying conditions should still be the correct wet surface temperature, that is, the temperature at which $\Delta$ in (5) should be evaluated to estimate the hypothetical wet surface evaporation rate [but note that $\Delta$ in (1) is always taken at air temperature]. During the regional drying process, Crago and Qualls (2021) showed that $e_a$ slides down the air isenthalp as drying progresses, while $T_0$ is found just as it would be for a saturated surface, namely using (12). So, use of (12) to determine $T_0$ for either saturated or unsaturated surfaces seems to have
good support, and this $T_0$ value can be used to predict wet surface evaporation rates from (5) with (4). The process is described above as a temporal drying of the region, but the analysis of Crago and Qualls (2021) is concerned only with the current status of the land and lower atmosphere, not with the drying or wetting pathway to that status.

With respect to the best formulation for the CR, there is unfortunately no consensus (e.g. Crago et al. 2022, Han and Tian, 2018). However, with FLUXNET data from Australia, Crago et al. (2022) found that the $y=X$ formulation was the best overall for predicting latent heat fluxes under many conditions. Given the wide range of methods represented in Figures 3 and 4, it is actually striking how little variation there is among the various methods with respect to RMSD (Figure 4; Tables 2 and 3).

Comparison of the four methods associated with hypotheses 1-4 suggests that all the hypotheses can give good estimates in many cases. In section 5.2 we will compare the methods in the context of an examination of these hypotheses.

**5.2 Examination of the hypotheses**

As discussed in section 1, the objective of this study is to evaluate different hypotheses or conceptualizations regarding $\alpha$, by using them to estimate $\alpha$ itself, actual wet surface evaporation (5), and hypothetical wet-surface evaporation as a part of a CR model that predicts actual regional evaporation rates (17). As discussed in section 5.1, outcomes from these conceptualizations are used to evaluate the hypotheses stated in the introduction. Including a range of hypotheses in this process makes it more likely that the correct conceptualization will be included and identified as the best.

Hypothesis 1, based on the $\alpha_c$-method, has been the default hypothesis in the majority of work with (5) and within CR formulations (e.g., Brutsaert, 2015, 2023, p. 148; Crago et al., 2016, 2022). Growing evidence that wet-surface, minimal-advection $\alpha$ actually has a fairly wide range of values (e.g., McNaughton and Spriggs, Lhomme, 1997 a,b, Raupach, 2000) might raise doubt regarding our ability to accurately estimate wet surface evaporation. Clearly, unexplained variability is a real challenge, but the $\alpha_c$ estimate performs quite well in predicting actual wet surface evaporation and in the CR model (Figures 3 and 4, and Tables 2 and 3).

A possible explanation for this surprisingly good performance begins with work by Szilagyi et al. (2014) and Andreas et al. (2013), who showed that much of the variability of $\alpha$ is due to temperature, with $\alpha$ increasing with decreasing temperature. Several formulations for $\alpha$ in terms of temperature are given in Figure 5, including $\alpha_A$, $\alpha_m$, $\alpha_{max}$ (defined in the Figure 5 caption), and a multi-term polynomial developed by Szilagyi et al. (2014) for $\alpha$ over saturated land surfaces. Because $\alpha$ is a function of temperature, it is likely that many of the large values of $\alpha_{ref}$ in Figure 3 (top panel) correspond to cold temperatures, which typically imply low available energy. Because available energy is small, relatively large errors in $\alpha$ result in only small absolute errors in wet surface evaporation using (5). Thus, the fact that the global constant value of $\alpha$ is too small for these low temperature sites does not result in large absolute errors in wet surface evaporation rates. Nevertheless, this is clearly not the best-supported of the four hypotheses.

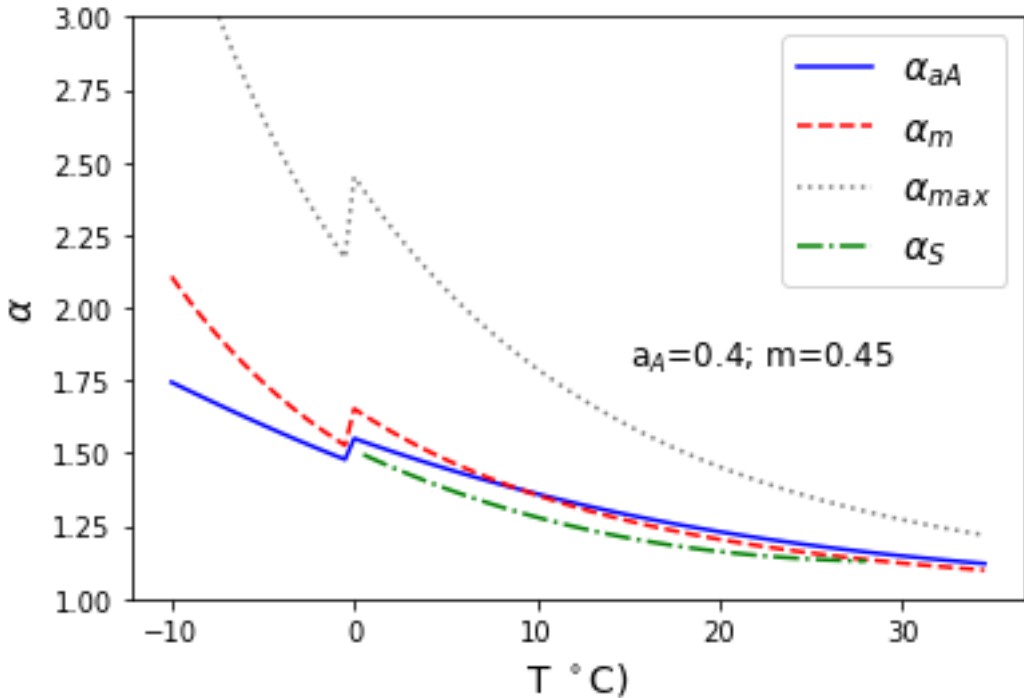

Figure 5. Variability of α from (8) and (9) with wet surface temperature. The value $a_A$=0.4 was chosen because it was recommended by Andreas (2013); the value of $m$=0.45 was chosen to approximately mimic the trends with the Andreas method. Here $\alpha_{max}$=1+γ/$\Delta_{T0}$, the maximum value of α suggested by Priestley and Taylor (1972). The green dash-dot line is $\alpha_S$, based on the third-order polynomial suggested by Szilagyi et al., 2014, on the basis of their analysis of ERA-Interim data over saturated surfaces.

Hypothesis 2, based on the $\alpha_A$-method, assumes a constant ratio ($a_A$) of the Bowen ratio between equilibrium and minimal-advection conditions. The resulting equation for α (8) is able to account for much of the systematic variability of $\alpha_{ref}$ due to temperature variability because of the variable $\Delta_{T0}$ in (8). Figure 3 (upper panel) shows that $\alpha_A$-estimates of α do increase as $\alpha_{ref}$ increases, but not as quickly as the reference values. While the trend is not matched perfectly, the $\alpha_A$-method is clearly an improvement over the $\alpha_c$-method in terms of predicting $\alpha_{ref}$. The method also performs well at predicting actual wet surface evaporation and actual evaporation (Figures 3 and 4), and it provides the smallest RMSD for estimating actual evaporation from (17) for both monthly (Table 2) and daily (Table 3) data. With a clear definition and consistently-good performance, hypothesis 2 has considerable support. However, it is not obvious (based on physical principles) why $a_A$ in (7) ought to be a constant. Overall, Hypothesis 2 gains considerable support from the data presented here.

Hypothesis 3, based on (9), also captures much of the variability of $\alpha_{\text{ref}}$. Equation (9) is correct to within the accuracy of Penman's (1948) well-known assumption regarding $\Delta$, provided that $RH$ is the actual measured relative humidity. When measured $RH$ is replaced with the parameter $RH$, (9) provides an estimate for $\alpha_{\text{ref}}$. Based on (1) combined with (5), (9) suggests that the optimal value of $RH$ should ideally represent the relative humidity that characterizes wet surface evaporation with minimal advection. Note that in (9), $\alpha$ depends on $f(u)$, $R_n$-$G$, and temperature.

As seen in Tables 2 and 3, the $RH$-method does not rank highly for prediction of $\alpha$, wet surface evaporation, or actual evaporation. Hypothesis 3 makes a very intuitive claim regarding wet-surface minimal-advection evaporation, namely, that it is associated with a particular value of relative humidity. While this method is conceptually appealing and it performs relatively well with some subsets of the data (not shown), its performance in this study is not as good as that of hypotheses 2 and 4. Thus, this study does not provide much support for hypothesis 3.

Hypothesis 4 assumes that minimal advection has been achieved when $\alpha$ is a specified fraction ($m$) of the distance from $\alpha=1$ to the maximum physically-realistic value of $\alpha_{\text{max}}=1+\gamma/\Delta_{\text{T0}}$ (Priestley and Taylor, 1972). The idea of $m$ being this fraction is clear and understandable, but it is not immediately obvious that it must be true on physical grounds. Overall, this method gives the lowest RMS error for estimating $\alpha$ and $LE_w$, it performs nearly as well as the $\alpha$A-method in estimating $LE$ using the CR (17), and it does this at both monthly and daily time scales. Furthermore, the other statistics included in Tables 2 and 3 are consistently favourable for this method. The data examined here seem to provide support for this method comparable to the $\alpha_A$-method.

Note that another hypothesis was considered for inclusion, based on the $\alpha_S$ curve, developed by Szilagyi et al. (2014) for saturated land surfaces and included in Figure 5. The fact that $\alpha$ is a strong function of temperature is an important insight. However, the valid temperature range of their curve is more limited (from 0 to 28 °C) than the temperatures in the dataset, and variability of $\alpha$ with temperature is already included in hypotheses 2, 3, and 4. Also, these hypotheses can be stated in terms of the parameters $a_A$, $RH$, and $m$, respectively, which have well-defined and physically-meaningful definitions. Therefore, no fifth hypothesis was evaluated.

Different data points might be assumed to represent wet surface conditions depending on the threshold value of $T$ as illustrated in Figure 1. But different points could also result for a given $T$ value for different values of $z_0$ used in the wind function (3). The data presented here have used the Wang et al. (2020) $z_0$ and $d_0$ formulations as described above. But eddy covariance measurements of friction velocity $u_*$ (m s$^{-1}$) are available for most of the sites and measurement periods. This means the logarithmic wind profile $u=(u_*/k)\log[(z_u-d_0)/z_0]$ (e.g., Brutsaert, 2023, p. 41) can be solved for $z_0$ for each measurement period for which $u_*$ is available. This value of $z_0$ is specific to a particular site and a particular month or day, so it accounts for

roughness variations with season and wind direction. With these data, the values of $z_0$ calculated in this way are somewhat smaller than those found with the Wang et al. (2020) formulation, which causes $LE_p$ values to be smaller and more data points to be identified as wet surface values. Nevertheless, a figure similar to Figure 1 but using these new $z_0$ values (not shown) suggests that $T=0.9$ is still appropriate. This method results in root mean square differences (see Supplement, Tables S2 and S3) comparable to those in Tables 2 and 3. The number of data points differ because not all time periods had $u^*$ measurements and because different $z_0$ values resulted in different data points qualifying as wet surface values. However, the key points remain unchanged. Results from the Wang et al. (2020) version of $z_0$ and $d_0$ are shown herein because they represent the way the roughness of land surfaces is usually estimated.

## 6 Conclusions

Four hypotheses regarding the Priestley and Taylor (1972) parameter α were considered. Each of them has a different assumption regarding the nature and variability of α. In the first hypothesis α is constant; in the second it represents a ratio of two Bowen ratios; in the third, it represents conditions at a given relative humidity value, and in the last, it can be seen as a midpoint between theoretical maximum and minimum values. Using FLUXNET data from a total of 171 stations, α, $LE_{PT}$, and actual evaporation values are compared to reference values in an attempt to determine which hypotheses best explain the data.

The second and fourth hypotheses generally produce the best results. In both of these, α is dependent on temperature, although the functional forms of the relationship are different. The third hypothesis has a very intuitive physical interpretation, but it tends not to work as well as the $\alpha_A$- and $m$-methods. But overall, the data in this study provide the most support for hypotheses 2, the $\alpha_A$-method, and 4, the $m$-method. According to hypothesis 2, the actual Bowen ratio under wet surface conditions with minimal advection is a constant fraction of the Bowen ratio under equilibrium conditions (4). According to hypothesis 4, α for wet surfaces remains at a constant fraction ($m$) of the distance between the minimum of value of one and the maximum value of $1+\gamma/\Delta_{T0}$. Since $\Delta_{T0}$ is a function of the wet surface temperature $T_0$, so is $\alpha_m$.

Without a need for any additional data, the temperature-dependence of α can be included in evaporation equations. It seems appropriate to include this dependence in applications of the Priestley-Taylor (1972) equation, and in particular in the use of CR models to estimate actual evaporation over drying surfaces. It is striking that four distinct hypotheses for how to understand the physical meaning of α can be stated clearly and that they have real implications regarding the nature and the numerical value of α.


## Acknowledgements

Support provided by the Ministry of Innovation and Technology of Hungary from the National Research, Development and Innovation Fund, financed under the TKP2021 funding scheme (project# BME-NVA-02) is kindly acknowledged.

## Data Availability

All data were downloaded from the FLUXNET website (http://fluxnet.org). Data pre-processing by FLUXNET staff follows procedures described by Pastorello et al. (2020). The datafiles and Python code (Wet_surface_evap_public.ipynb) have been uploaded to Zenodo with doi:10.5281/zenodo.8172604, URL: https://doi.org/10.5281/zenodo.8172604. Instructions are given in the description at Zenodo.


The project is also available from GitHub at https://github.com/r-crago/FLUXNET_Wet_sfc_evap.

## Author Contributions

R. Crago: Conception of study; acquisition and analysis of data; rough draft, edits

R. Qualls: Further development of concepts; edits

J. Szilagyi: Further development of concepts; edits

## Competing interests

The authors have no competing interests.

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
