# Peer review of "What is the Priestley-Taylor Wet-Surface Evaporation Parameter? Testing Four Hypotheses"

_EGUsphere, 2022_

## Referee Comment (RC1)

**Review of EGUsphere manuscript # 2022-217**

**Title: What is the Priestley-Taylor Wet-Surface Evaporation Parameter? Testing Four Hypotheses**

The manuscript comprehensively evaluates different approaches about the Priestly-Taylor $\alpha$ parameter. Considering the broad application of P-T equation in quantification of evaporative fluxes at different scales, the study can potentially provide insights for hydrology and climate modeling approaches. However, there are some aspects in the study that need clarification or should be addressed by the authors.

- Acknowledging that the definitions of wet surface and potential evaporation are always controversial as reflected in different studies, I think the description in line 56 tacitly ignores the key role of atmospheric forcing and its coupling with land surface; in other words, if the surface was truly saturated then the measured meteorology would have been different.

- In the present structure of the manuscript and representation of section 2.2, it is a bit difficult to follow the linkage of the work with CR approach (of course a direct application of Priestly-Taylor $\alpha$ parameter is there). As already reflected in the title, the manuscript has a very clear objective. Perhaps rewriting of that section and better representation of the results around Figure 4 will improve the coherency of the manuscript.

- Eq. (6) and line 167: I think the $\leq$ sign on the RHS is not intuitive here. Air is already saturated at equilibrium condition; $\alpha=1+\gamma/\Delta$ implies $LE=Rn\text{-}G$; this means $H$ is zero or $Ta=Ts$. Since both surface and air are already saturated (i.e., $RH=1$), this means vapor pressure gradient suddenly approaches zero and evaporation stops!

- Line 128: simply even at equilibrium condition, part of available energy will be exchanged via sensible heat flux.

- Line 188: surface drying within the CR approach implies drier and "warmer" near surface air as part of available radiative energy not used by $LE$ is now released in form of sensible heat flux.

- Line 35: is not it Bouchet (1963)?

- Line 64: it might be helpful to elaborate more on the objectives in Crago and Qualls (2013) and clarify differences.

- In general, the quality of figures and representation of information (especially statistical attributes in legends) remains a bit low. Captions can benefit more from direct interpretation of the results.

---

## Author Comment (AC1)

Response to Reviewer 1

We thank Reviewer 1 for helpful comments. Reviewer 1's comments are in black, and our responses are in red:

The manuscript comprehensively evaluates different approaches about the Priestly-Taylor α parameter. Considering the broad application of P-T equation in quantification of evaporative fluxes at different scales, the study can potentially provide insights for hydrology and climate modeling approaches. However, there are some aspects in the study that need clarification or should be addressed by the authors.

- Acknowledging that the definitions of wet surface and potential evaporation are always controversial as reflected in different studies, I think the description in line 56 tacitly ignores the key role of atmospheric forcing and its coupling with land surface; in other words, if the surface was truly saturated then the measured meteorology would have been different.

This manuscript is primarily looking at evaporation from surfaces we are (relatively) confident are actually wet. Figure 1, Table 1, and Figure 3 apply to such wet surfaces. The hypothetical wet surface evaporation comes in when using the Complementary Relationship to calculate evaporation from a wide range of surfaces in Figure 4.

When asking "What would happen if conditions were the same as they are, except…..?", it is important to define what stays the same. Most interpretations of the Priestley-Taylor equation assume the available energy stays the same and the air temperature stays the same (i.e., Brutsaert, 1982, 2005, 2015).

However, Szilagyi and Jozsa (2008) and Szilagyi and Schepers (2014) give good arguments for the idea that a small wet patch also maintains the same skin temperature while the region dries out, as long as the available energy and the wind speed are held constant. Szilagyi, Crago and Qualls (2016) argue that this wet patch skin temperature should be used to determine the saturation vapor pressure slope ($\Delta$) when estimating equilibrium evaporation.

This hypothetical saturated surface skin temperature is crucial for estimating equilibrium evaporation. Qualls and Crago (2020), following Philip (1987) note that equilibrium evaporation is simply the smallest possible evaporation rate from a saturated surface for a specified available energy and surface temperature. In a graph of vapor pressure versus temperature, Philip (1987) showed that wet surface evaporation is at its lowest when a line from $[T_0, e^*(T_0)]$ to $(T_a, e_a)$ is tangent to the saturation vapor pressure curve at wet patch skin temperature $T_0$.

Putting this together, the hypothetical wet surface temperature $T_0$ of a drying surface can be estimated using (12), and equilibrium evaporation can be estimated from that $T_0$ in addition to $(R_n$-$G)$. Thus, there are good reasons to think that real wet surface evaporation would be similar to our hypothetical values. Alternative views are discussed in:

Tu and Yang, 2022, On the estimation of potential evaporation under wet and dry conditions, WRR 58(4), https://doi.org/10.1029/2021WR03148.

Szilagyi, 2022, Comment on "On the estimation of potential evaporation under wet and dry conditions, WRR, https://doi.org/10.1029/2022WR033264.

Yang, Tu, and Roderick, 2022, Reply to Comment on "On the estimation of potential evaporation under wet and dry conditions, WRR, https://doi.org/10.1029/2022WR033674.

- In the present structure of the manuscript and representation of section 2.2, it is a bit difficult to follow the linkage of the work with CR approach (of course a direct application of Priestly-Taylor α parameter is there). As already reflected in the title, the manuscript has a very clear objective. Perhaps rewriting of that section and better representation of the results around Figure 4 will improve the coherency of the manuscript.

To help make this connection, at the end of the first paragraph of section 2.2, we plan to add: "Thus, an estimate of the Priestley-Taylor α is an integral part of the CR, and the performance of CR models making use of the four different hypotheses regarding α can serve as a further test of the hypotheses."

- Eq. (6) and line 167: I think the ≤ sign on the RHS is not intuitive here. Air is already saturated at equilibrium condition; α=1+γ/Δ implies LE=Rn-G; this means H is zero or Ta=Ts. Since both surface and air are already saturated (i.e., RH=1), this means vapor pressure gradient suddenly approaches zero and evaporation stops!

Just to clarify, equilibrium conditions refer to α=1, not to α=1+γ/Δ. Regarding α=1+γ/Δ, H=0 implies that potential temperature is constant with z, rather than actual temperature, so there can be a small e-gradient even when H=0. However, we have changed from ≤ to simply <, since the point is to establish that α cannot go higher than 1+ γ/Δ.

When α > 1+ γ/Δ, it implies that H<0. The reasoning for this limit was given by Priestley and Taylor (1972) : "In the absence of advection…..it is unlikely that an inversion will prevail so as to make H negative." That is, over a period of a day or longer, it is unlikely that H<0 while LE>0 unless there is strong advection warming and drying the air above the wet surface. An example would be warm and dry continental air blowing across a cool ocean, violating the "absence of advection" criterion (Priestley and Taylor, 1972).

- Line 128: simply even at equilibrium condition, part of available energy will be exchanged via sensible heat flux.

Yes, there will be positive sensible heat flux throughout the range 1 ≤ α < (1+ γ/Δ). On this line of the manuscript we are trying to explain WHY there will be some sensible heat flux.

- Line 188: surface drying within the CR approach implies drier and "warmer" near surface air as part of available radiative energy not used by LE is now released in form of sensible heat flux.

Yes. There are two sides of the same coin. Our explanation focuses on the water availability while yours focuses on how the available energy is allocated.

- Line 35: is not it Bouchet (1963)?

Yes. We will correct the reference.

\- Line 64: it might be helpful to elaborate more on the objectives in Crago and Qualls (2013) and clarify differences.

We plan to change the wording to: "The objective here is to gain a better conceptual understanding of alpha (c.f., Crago and Qualls, 2013)."

\- In general, the quality of figures and representation of information (especially statistical attributes in legends) remains a bit low. Captions can benefit more from direct interpretation of the results.

We plan to modify the graphs so that the text in the graphs is easier to read.

---

## Author Comment (AC2)

We thank Reviewer 2 for helpful comments. Reviewer 2's comments are in black, and our response is in red:

Reviewer 2

This manuscript comparatively tests four hypotheses of the Priestley-Tyalor wet-surface evaporation and calculated the corresponding parameters. It is an interesting work for the research on evaporation, from both the theoretical and application perspetives. I think it is worth for publishing after addressing several comments below.

Major comments:

1.   The criterion of LEref>0.9LEp for wet surface conditions requires acurate wind function f(u) for LEp. The actual wind function may vary with the aerodynamic conditions, the boundary layer characteristics, or even the magnitude of wind speed. The wind function (3) with the fixed canopy height used in this study may derivate from the actual one (Han et al., 2021), especially with the growth of the vegetation. Let's write Ep with fixed wind function (3) as Ep'. E=alpha*Ee is equivalent to E/Ep'= alpha*Ee/Ep'. Then, E/Ep' may be substantially less than 0.9 by using wind function (3) with fixed canopy height, and substantial data which should be taken as under wet surface conditions may be excluded. Under this conditions, the RH may be limited to large values artificially to make sure that Ee/Ep'>0.9. So, an evaluation on the result of the chosen for near wet surface conditions is needed, against other methods, or on the real wet surfaces, such as wetlands. What is the proportion of data left for a permanent wetlands by this criterion with the fixed wind function?

The criterion $LE_{ref}$>(some fraction of $LE_p$) seems to us to be the most straightforward way of expressing the wet surface condition. For a wet surface, the equation for the actual evaporation is literally the same as the equation for $LE_p$, so if $LE$ is approximately $LE_p$, the surface should be wet. We agree that the wind function can play a role, and that there is some uncertainty in the values of $z_0$ used, where $z_0$ is given by $h/8$ and h is the canopy height.

To address this issue, we plan to add the following at the end of section 5:

Different data points might be assumed to represent wet surface conditions depending on the threshold value of $T$ as illustrated in Figure 1. But different points could also result for a given $T$ value for different values of $z_0$ used in the wind function (3). Han et al. (2021) noted that different assumptions regarding the proper wind function parameters (e.g., $z_0$) can produce disparate results in wet surface evaporation studies. The data presented here have used the Wang et al. (2020) $z_0$ and $d_0$ formulations as described above. But eddy covariance measurements of friction velocity u$^*$ (m s$^{-1}$) are available for most of the sites and measurement periods. This means the logarithmic wind profile $\underline{u}$=(u$_*$/k)log[($z_u$-$d_0$)/$z_0$] (Brutsaert, 2005) can be solved for $z_0$ for each measurement period for which $u_*$ is available. This value of $z_0$ is specific to a particular site and a particular month or day, so it accounts for roughness variations with season and wind direction.

With these data, the values of $z_0$ calculated in this way are somewhat smaller than those found with the Wang et al. (2020) formulation, which causes $LE_p$ values to be smaller. Nevertheless, a figure similar to Figure 1 but using these new $z_0$ values (not shown) suggests that $T=0.9$ is still appropriate. This method results in root mean square errors comparable to those in Tables 1 and 2 (see Supplement, Tables S.2.1 and S.2.2). The number of data points differ because not all time periods had $u^*$ measurements and because different $z_0$ values resulted in different data points qualifying as wet surface values. However, the key findings remain unchanged. Results from the Wang et al. (2020) version of $z_0$ and $d_0$ are shown because they represent the way the roughness of land surfaces is usually estimated.

Finding the fraction of permanent wetlands that meet our criterion is a good idea. However, the "permanent wetland" classification is the only wetland category used by IGBP, and some seasonal wetlands are placed in this category. For example, the Fogg Dam (AU-Fog) description at https://www.ozflux.org.au/monitoringsites/foggdam/foggdam_description.html says, "the flux tower site was classified as a seasonally flooded wetland." Thus, not all evaporation from these sites should be expected to be wet surface values. Of the 323 months of data from "WET" sites, 105 months were classified as wet surface evaporation. We believe our threshold value is a more reliable way of identifying wet surfaces.

2. The result of the third hypothesis with large values of RH near the unity (Table 2 and 3) may be affected by above data chosen method, as Ee/Ep'>0.9 requires large RH.

The criterion is actually $LE_{ref}/LE_p>0.9$. All the methods are susceptible to bias if errors are made in determining which measurements represent wet surfaces. That is why various thresholds of $T$ and different estimates of $z_0$ (see response 1 above) have been considered.

3. For the hypothesis 4. Are the days of months with negative Href were excluded? Then, the data outside the range of Eq. (6) were excluded. The results may be influenced by this.

Yes, averaging periods with negative $R_n$, $R_n$-$G$, $H$, or $LE$ were all excluded. This was done Following Andreas et all. (2014) who placed wet surface evaporation scenarios into several categories. The case where H and LE are both positive is the one of interest in this study. See also Priestley and Taylor (1972).

4. Table 2 and 3 only supply the optimized parameter of the other three hypotheses. How the calculated alpha varies? Are the mean or median values related with ac?

The paper is intended to test whether $a_A$, $RH$, and m are more fundamental than alpha. While the resulting average values of alpha from all these methods will be similar, Figure 3 shows that the different methods produce different individual values of alpha; these individual values are our interest here.

5.  Line 397: The Priestley-Taylor coefficient was not regarded a constant in Han and Tian (2018), but with seasonal variations, to the best of my knowledge. Please refer to Han et al., (2021).

Thank you. We have included Han et al. (2021) in the literature review now, and Han and Tian (2018, 2020) were removed from the sentence on line 397.

6.  Lines 400-410. The performance with constant ac is good by considering all the data. But bias exist under the conditions with small values or large values of LEref, as shown in Figure 3. Is it possible to give some discussion?

The point of writing this paper is that the constant-α model needs to be re-considered. So, we are not surprised to find the bias pointed out by the reviewer. In a way, this makes sense, because for a wet surface, we expect $LE = LE_p = \alpha LE_e$ (Brutsaert, 2005). But $LE_p$ from (1) has two terms, $LE_e$ (4) and the second term, which we will call $LE_{aero}$ (e.g., Han and Tian, 2018), where $LE_{aero}$ depends mostly on the vapor pressure deficit and the wind function. This gives:

$$LE_e + LE_{aero} = \alpha LE_e$$

Or

$$\alpha = 1 + LE_{aero}/LE_e$$

Supposing $LE_{aero}$ varies somewhat independently of $LE_e$, small values of $LE_e$ would tend to result in large values of α and vice versa (see Han et al. 2021).

Other comments:

1.  Page 16 and 17, Typo for Table 2 and 3. Tables will be corrected with the latest computed values

2.  Table 2. The intercept of RH with optimized LE is 15.54, but 15.52 in Figure 4. Table will be corrected with the latest computed values.

3.  Line 342: four hypotheses? Yes, thank you, we will correct it.

Reference:

Han, S., Tian, F., Wang, W., & Wang, L. (2021). Sigmoid generalized complementary equation for evaporation over wet surfaces: A nonlinear modification of the Priestley–Taylor equation. Water Resources Research, 57(9), e2020WR028737. doi:10.1029/2020wr028737

---

## Author Response (AR1)

Dear Editor,
Thank you for the careful review of the manuscript and your substantive comments. We have done significant new analyses, and have thoroughly revised the manuscript in response to your comments as well as those of the two reviewers. We believe the manuscript is much improved. Our Responses are in red.

Regards,
R. Crago
J. Szilagyi
R. Qualls

25 Feb 2023
**Editor decision: Reconsider after major revisions (further review by editor and referees)**
by Stan Schymanski
**Public justification (visible to the public if the article is accepted and published)**:
Dear authors,
Thank you for the detailed responses to the referee comments. Both referees agree that the manuscript is potentially publishable in HESS, but request major revisions, due to lack of clarity and methodological questions. Upon re-reading the manuscript, I agree about the lack of clarity, and I am left with a lot of open questions about the methods and doubts about the insights provided by this manuscript. Therefore, I believe that the manuscript requires a thorough overhaul to improve clarity and potentially eliminate methodological flaws, as pointed out by the referees and in my own comments and questions below.
It would be very helpful if you could accompany the revised manuscript by a document where you explain the modifications in response to the points raised by the referees and myself. Thank you in advance.

GENERAL COMMENTS
In the manuscript, you analysed daily and monthly fluxes, stating that this is what the equations are mostly used for. But since you are citing Slatyer and McIlroy (1961), you may give some consideration to their statement on P.58: "However, use of average values in equations like 3.13, containing products of variable quantities, means neglecting the effects of short-term correlations between these quantities, which could sometimes lead to significant errors (cf. discussion of eddy flux versus mass flux, Section 9(f)). For this reason individual values of all quantities, including G wherever possible, should be entered into the equation as frequently as they can be obtained, and only the consequent
values of E should be averaged."
Since the eddy covariance data does contain hourly data, it would be worthwhile following their advice or at least explaining why you did not use it.

In Section 3.1, we added the following paragraph to address this comment:

The CR has been used at time scales from hourly to yearly (Brutsaert, 2023, p. 147), but the CR, the Penman equation (1), and the Priestley-Taylor equation (5) are most typically applied to daily to monthly average values (McMahan, 2013). These equations assume adjustment of atmospheric temperature and humidity to the energy fluxes at the surface, and adjustment of the surface to conditions in the lower

atmosphere (Brutsaert, 2023, p. 147), and the diurnal cycle makes this adjustment unlikely over periods less than 24 hours (McMahon, 2013). While the CR could potentially be applied to hourly data, such that the daily CR estimate $LE_{est}$ would be the average of the hourly estimates, the approach here is to use daily (monthly) average input values to produce daily (monthly) energy fluxes. This approach recognizes that hydrological processes often apply at a limited range of temporal and spatial scales. For example, infiltration excess runoff applies at point scales while saturation excess applies to the watershed scale (Chow et al., 1988). Here, the CR applies at regional spatial scales and daily to monthly temporal scales.

I do not understand why new parameters are chosen for the use with the CR. Actually, I struggled to understand the procedure, as explained in the detailed comments. Here is what I took away after re-reading multiple times: First you fit parameters to reproduce LE_ref as closely as possible for different filter settings (closeness of LE_ref and LE_p). Then you compare the resulting alpha values with LE_ref/LE_e and the estimated LE_PT with LE_ref (Fig. 3). The differences between methods are minimal here. Then, you use the PT-equation in combination with the CR, but re-calibrate the parameters. At some stage, I assume when you use the CR, you seem to have calibrated the parameters to either make alpha match LE_ref/LE_e as closely as possible, or to make LE_PT match LE_w as closely as possible or to make LE_PT match LE_ref as closely as possible (Tables 1 and 2). But I am really guessing here.

We have made many changes to section 3.1 to emphasize exactly what procedures we followed. Much of the confusion seems to have been with regard to different parameter values for the different steps of the process. Specifically, tuned values for each of $\alpha_c$, $\alpha_A$, RH, and m are determined three times: 1) to find best fit $\alpha_{est}$; 2) to find best fit $LE_{PT}$, and 3) to find best fit $LE_{est}$ using the CR (17).

Regarding the need for multiple tunings and what the significance of the different tuned values is, we have included in Section 5.1 the following two paragraphs:

> Different tuned parameter values were found to calculate $\alpha_{est}$ (Figure 3, top panel) $LE_{PT}$ (Figure 3, bottom panel), and $LE_{est}$ (Figure 4). Ideally, the tuned parameter values would remain nearly identical in the three cases. A likely explanation for this difference is as follows: In the top panel of Figure 3, all $\alpha_{ref}$ values count equally in determination of tuned parameter values that produce the best-fit $\alpha_{est}$ for each of the methods. But in the bottom panel of Figure 3, $\alpha_{est}$ values that correspond to small values of $(R_n-G)$ have far less influence on the RMSD of $LE_{PT}$ than those corresponding to larger $R_n-G$. If $R_n-G=5$ W m$^{-2}$, an increase in alpha from 1.1 to 1.3 only increases $\alpha(R_n-G)$ from 5.5 to 6.5 W m$^{-2}$, whereas if $R_n-G$ is 200 W m$^{-2}$, it increases from 220 to 260 W m$^{-2}$, so that the larger $R_n-G$ would influence RMSD of $LE_{PT}$ more. Similarly, when moving to the CR estimate $LE_{est}$ in Figure 4, the CR estimates again apply different weight to the various estimates of $\alpha$, so that different tuned parameter values result here as well. Actually, Brutsaert (2023) treats the parameter $\alpha$ in (5) as a completely-different parameter from the $\alpha$ embedded in (17). While the present authors consider both to be the same parameter, we recognize that its tuned value could vary depending on the context.

> For reference, when the parameter values used in Figure 3 (lower panel) are used in the CR (namely $\alpha=1.29$ for the $\alpha_c$-method; $\alpha_A=0.31$ for the $\alpha_A$-method, $RH=0.76$ for the $RH$-method, and

*m*=0.58 for the *m*-method), RMS errors increased (from 19.37 to 20.12; from 18.61 to 19.25; from 21.13 to 26.84; from 18.65 to 19.29, respectively). Note that the $\alpha_A$- and *m*-methods still provide the lowest RMSD values.

In Section 4, the following clarification is now included:

As described in section 3.2, using only wet-surface measurements, tuned values of $\alpha_c$, $a_A$, *RH*, and *m* were found that minimized RMSD between reference and estimated values of $\alpha$. Different tuned values of these parameters were also found for use in estimating wet surface evaporation using the Priestley-Taylor equation (5). Results for estimating $\alpha$ under wet-surface conditions are found in panel a of Figure 3, and results for estimating wet surface evaporation itself are shown in panel b; both sets of results are also provided in Table 2. Finally, still-different tuned values of $\alpha_c$, $a_A$, *RH*, and *m* were those which produced the minimum values of RMSD between $LE_{est}$ and $LE_{ref}$ using the CR formulation (17). That is, $LE_{est}$ is found by taking *y* found with (17), where $LE_w$ is given by (5), $LE_p$ by (1) through (3), and *y* by (13) through (17). This *y* was multiplied by $LE_p$ from (1) to get $LE_{est}$. The tuned parameters that result when the goal is to obtain the best fit between $\alpha_{est}$ and $\alpha_{ref}$, are different than those when the goal is to obtain the best fit between wet-surface $LE_{PT}$ and $LE_{ref}$, and in turn these are different than when the goal is to obtain best fit between $LE_{est}$ and $LE_{ref}$ using (17). Reasons and implications for these differences will be discussed in section 5.

Since you are filtering for data points where LE_ref is very close to LE_p, it is not clear to me what is the use of the CR, as the CR is supposed to be relevant for estimating LE_ref when it is not close to LE_p. Secondly, since you are re-calibrating the parameters, what is the predictive power of the method? If the parameters need re-calibration before being used in the CR, this suggests to me that a well calibrated LE_PT is not useful in the CR and hence the CR as you used it, is not useful. I think it would be very helpful if you could describe at the beginning of the Methods section the overall approach (what is fit to achieve what) and what you expect to see.

This confusion probably started in section 3. The first two parameter tunings (to find best fit values of αest and of LEPT) use only wet surface data. When we move to the CR we use all the available data (wet surface and non-wet surface). This has been clarified. With respect to the need for multiple calibrations, please see our response to the previous question. In addition, the CR method requires the use of hypothetical wet surface evaporation, therefore this part of the analysis incorporates an assessment of the wet surface alpha by means of its use in the CR.

In Tables 2 and 3, very different parameter values were needed to reproduce either alpha, or LE_w or LE. The text is not very clear about how this was done (see detailed comments), but also not what it means. I had the impression that the different methods did not matter too much when matching LE_PT to LE_ref, so why do they matter when using the CR? Your explanation in L362-365 is not very clear. Also, in the text, you describe the RMSE as the deviations between the simulated values and the reference, but in the figures and tables

you provide it along with the parameters of the linear fit (slope and intercept), so I am not sure if the RMSE was not calculated based on the deviations between the simulated data and the regression line. For example, in Fig. 4 top left, the reported "RMS" is very close to that of the top right panel, which has a similar spread of data, but a much smaller regression slope and therefore probably greater deviations from the 1:1 line.

Once again, sections 3.2, 3.3, and 3.4 have been substantially re-written to make clear why there are different parameter values. In these sections there is a clear explanation that the best fit values are found by minimizing RMSD(estimate, reference value), where the difference is simply the estimate value minus the reference value.

Referee 1 criticised that your manuscript ignores surface-atmosphere feedback when discussing the difference between wet surface and potential evaporation. Although most of your analysis focuses on wet surfaces, where both are very similar, as you state in your response, when you relate to the CR method, it would be important to explain the feedback between evaporation and atmospheric conditions, mainly temperature and humidity. The temperature feedback is also not mentioned in L188. In your response and in some of the text, you focus a lot on surface temperature, but I really do not see the point of re-introducing surface temperature into a framework formulated explicitly to eliminate the need for specifying surface temperature (Penman, 1948). In this respect, I have noticed a confusion about the meaning of Delta, as Penman did explicitly use the slope at air temperature, not at surface temperature. Since the eddy covariance data does not contain surface temperatures, it is actually not clear how you calculated Delta in this study.

In Section 2.2, we include the following description of the feedback:

> In the Complementary Relationship (CR) between actual and potential evaporation (Bouchet, 1963), regional evaporation from a saturated surface, the apparent-potential evaporation rate, and the actual evaporation rate are all identical (Brutsaert, 2015). According to the advection-aridity approach (Brutsaert and Stricker, 1979), apparent potential evaporation corresponds to the Penman equation (1) or to the evaporation from a small wet patch, and the wet regional surface rate corresponds to the Priestley and Taylor (1972) equation (5). As the surface dries, less water is available to evaporate, so actual evaporation decreases. This results in a drier lower atmosphere, which increases apparent potential (wet patch) evaporation. Conversely, if the lower atmosphere becomes dry (in the absence of significant dry advection), this implies that regional evaporation rates are low. Thus, evaporation and apparent potential evaporation change in opposite directions—they complement each other.

Regarding the wet surface temperature, we include the following

> Figure 3 and Tables 2 and 3 provide evidence that all four methods provide acceptable estimates of actual wet surface evaporation rates. But what about estimation of hypothetical wet surface evaporation rates? Szilagyi and Jozsa (2008) and Szilagyi and Schepers (2014) have provided good evidence that a small wet patch within a drying region with wind speed and available energy held constant should maintain a constant surface temperature. The use of (12) to get the temperature of an actually-saturated region is straightforward. Crago and Qualls (2021), Qualls and Crago (2020) and Szilagyi (2021) show graphically, using $e$ versus $T$ graphs, how air and ground-surface isenthalps [lines of constant available energy on a $(T, e)$ graph] can be

determined, and they show that $T_0$ is simply the intersection of the ground-surface isenthalp with the saturation vapor-pressure curve.

There are three distinct explanations for why a best-fit estimate $LE_{est}$ using the CR (17) might differ from the $LE_{ref}$ values: First, α may not be estimated correctly (see the preceding paragraphs); second, the $LE_{PT}$ estimate may not adequately represent the hypothetical wet-surface evaporation rate; and third, the CR formulation may be inadequate. No method to distinguish the effects of these is apparent, so that the results in Figure 4 and Tables 2 and 3 provide only an indirect test of the adequacy of the four methods to estimate α for hypothetical wet surfaces.

Under drying surface conditions, since the wet surface temperature remains constant during drying (assuming $R_n$-$G$ and wind speed are constant—see Szilagyi and Schepers, 2014), $T_0$ found with (12) under drying conditions should still be the correct wet surface temperature, that is, the temperature at which Δ in (5) should be evaluated to estimate the hypothetical wet surface evaporation rate [but note that Δ in (1) is always taken at air temperature]. During the regional drying process, Crago and Qualls (2021) showed that $e_a$ slides down the air isenthalp as drying progresses, while $T_0$ is found just as it would be for a saturated surface, namely using (12). So, use of (12) to determine $T_0$ for either saturated or unsaturated surfaces seems to have good support, and this $T_0$ value can be used to predict wet surface evaporation rates from (5) with (4). The process is described above as a temporal drying of the region, but the analysis of Crago and Qualls (2021) is concerned only with the current status of the land and lower atmosphere, not with the drying or wetting pathway to that status.

Referee 1 also found Section 2.2 difficult to follow. Perhaps it would help to remove all the equations that are not actually used, and instead explain the concept a bit better (perhaps including a plot of the key CR formulation) and refer to the papers for the other equations. We have omitted Brutsaert's (2015) equation (the previous equation 13), and try to put the CR work into context. Specifically, we want to know if methods to estimate alpha or to estimate wet surface evaporation, also improve estimates of actual evaporation under non-wet conditions. That is, do wet-surface evaporation equations work to give *hypothetical* wet surface evaporation rates? This cannot be directly measured, so we indirectly measure it by comparing CR estimates of LE with reference values, under both wet and dry conditions.

Referee 2 is concerned about the effect of an inadequate wind function on your results and you mention additional analysis in your response. Could you include in the paper or SI the figure you mention in your response?

We decided to use tables similar to tables 2 and 3, except using z0 values derived for each day (month) based on the measured ustar values. That is, every site has a different value of z0 for each day (month) in the dataset. The major trends and conclusions are similar for this

analysis and for the analysis in the paper itself. This aspect of the work is discussed in Section 5.1:

> For reference, when the parameter values used in Figure 3 (lower panel) are used in the CR (namely α=1.29 for the $α_c$-method; $α_A$=0.31 for the $α_A$-method, $RH$=0.76 for the $RH$-method, and $m$=0.58 for the $m$-method), RMS errors increased (from 19.37 to 20.12; from 18.61 to 19.25; from 21.13 to 26.84; from 18.65 to 19.29, respectively). Note that the $α_A$- and $m$-methods still provide the lowest RMSD values.

SPECIFIC COMMENTS

L120-: Please provide page numbers when referring to books as sources. Slatyer and McIlroy (1961) based their derivations on Penman (1948), where Delta (s in Eq. 3.19 in SM1961) is evaluated at air temperature, not at the wet surface temperature, as stated in L121, and then again in L144. I am aware that Yang and Roderick (2019) also used Ts, but it would be worth pointing out here that Penman (1948) did not.

We have included page numbers for books. At equation (1), Penman's equation, we explicitly point out that delta is taken at the air temperature.

L131: I have not found that PT (1972) actually referred to Eq. 4 at all. They introduce alpha ad-hoc in their Eq. 5, so, as intuitive as it sounds, your statement may be misleading.

Agreed, we have reworded that section.

L264-: I do not understand this paragraph. What is the "realistic range of values" for each parameter, and how many values were randomly chosen? What does L266 mean? How was LE_e calculated? How is T0 obtained from the data for calculating the slope "at the wet surface temperature" in Eqs. 8, 9 and 11?

In Section 3.2, we provide more information about the range of the parameters:

> Trial values of the parameters $α_c$, $a_A$, $RH$, and $m$, where $α_c$ is a constant (global) value of α were selected randomly from a range of reasonable values (that is, α ranged from 1 to 1.6 and $a_A$, $RH$, and $m$ all varied from 0 to 1). These values were used at all sites and times satisfying the wet surface condition. Three thousand values drawn randomly from these ranges were evaluated to determine the optimal parameter values. These optimal parameter values were used to estimate different versions of $α_{est}$, namely $α_c$, $a_A$ from (8); $α_{RH}$ from (9); and $α_m$ from (11). These $α_{est}$ values were compared to $α_{ref}=LE_{ref}/LE_e$. The trial parameter values that minimized the root mean square difference (RMSD) between $α_{est}$ and $α_{ref}$ were taken to the be tuned parameter values; the values of $α_{est}$ will be called the best-fit values.

Just below (12) we noted that,

> "The wet surface temperature in (12) is $T_0$, which can be easily found from (12) with a numerical root finder. Equation (12), thus solved provides the wet surface temperature $T_0$ from data taken from either saturated or unsaturated surfaces (Szilagyi and Schepers, 2014)"

L274-: Why were again new parameter values fitted for the use in the CR?

Please see explanation above

L283: This would be better off in the figure caption instead of the main text.

This discussion of information contained in the text of figures has been moved to the figure caption.

L356 and throughout: I find the use of the word "optimize" very confusing here. It seems that you are referring to the target variable you want to reproduce, as the "optimized" variable. Most readers would associate with "optimized" variable the variable that is being tuned. Why not write "parameter values that reproduce observed alpha" instead of "parameter values that optimize estimation of alpha"? And then you could refer to the variables that you actually tuned as either "tuned" or "calibrated".
We've adopted the term "tuned" to refer to parameter values that provide the "best fit" of estimates to reference values, and the "best fit" is defined to be the smallest RMSD value.

L360-: If the parameters have to be re-calibrated depending on the context, this would suggest to me that they actually don't have a consistent meaning. Why was the CR not applied using the parameters obtained in the previous step?

We have now addressed this below your second comment.

L362-: This paragraph is hard to understand. What do you mean by "estimated correctly"? What is a "correct" value of alpha? What do you mean by "the LE_PT estimate LE_est may not adequately represent..."?

This section has been reworded

L372: This is the first time you mention the term "iso-enthalp". It would be good to explain what it means.

Yes, we've changed it to "isenthalp" and defined it as a line of constant available energy on a (T, e) graph.

L384: The RMS errors are similar, but the slopes of the correlation are quite different. It is actually not clear whether RMSE refers to the fitted curve or to the 1:1 line. In the former case, RMSE does not say much about how well one variable reproduces the other.

RMS is based on (reference – estimate)^2. That is, it refers to the 1:1 line.

Tables 1-2: Please describe RMSE, R, S, I, NSE. What do you mean by "Optimized variable"? Actually calibrated variable or the variable to be reproduced? What do you mean by LE_w and LE? I thought that LE_ref was observed LE filtered for wet conditions, so not sure what is the difference here. You also never mentioned the NSE in the methods, so this needs to be added.

Please see our answer 5 responses up. Also, RMSD, R, S., I. and NSE are all defined clearly in the Figure 1 caption, Table 1 footnotes, Figure 3 caption, and Figure 4 caption.

Fig. 1: The black lines are not visible. The caption does not explain the difference between plots in each column. Where are the reference values stated?

The graphs have been re-plotted and the caption explains the different plots

Fig. 3: What does alpha_est and alpha_ref refer to? Please mention that S is slope, I is intercept.
The text now explains that alpha_est is values of alpha estimated from one of the "methods" or hypotheses, and alpha_ref is LEref/LEe.

Fig. 4.: Please label the sub-plots as a, b, c, and d. Please also specifiy whether the RMS (I would call it RMSE) refers to deviations from the regression line or from the 1:1 line. The regression lines should also be visible.
The subplots have been labeled. The RMSD is now fully explained.

---

## Author Response (AR2)

Dear Stan,

Thanks for this opportunity to address a few remaining issues with this paper. Our responses are given in red.

Dear authors,
Thanks again for the careful revision of the manuscript and detailed responses to referee comments. The referees are both supporting publication of your article now, subject to technical corrections. I would still like you to consider/address a few last points before publication, as listed below. Thanks again for your patience and commitment to high quality research.

Best regards,
Stan

L193: In your response about surface-atmosphere feedback, you mention humidity, but not temperature, as suggested by the referee. Could you add this to your consideration?

Yes, thank you. The two sentences now read: "This results in a drier and warmer lower atmosphere, which increases apparent potential (wet patch) evaporation. Conversely, if the lower atmosphere becomes dry and warm (in the absence of significant dry advection), this implies that regional evaporation rates are low."

L261: Should be McMahon.

Thank you for catching this. It has been changed (see text two responses down).

L262: The Penman equation is not based on any assumptions about adjustments of atmosheric temperature and humidity, please correct. Could you also mention that Slatyer and McIlroy (1961) advised against such averaging: Slatyer and McIlroy (1961, P. 3-38): "However, use of average values in equations like 3.13, containing products of variable quantities, means neglecting the effects of short-term correlations between these quantities, which could sometimes lead to significant errors".

Thank you. This point and the next all refer to the same paragraph which we have reworked considerably, and provide below the next point.

L264-266: The reader would benefit a lot from a comparison between values obtained using hourly and monthly data. Here it sounds as if the results would be similar, but this is in contrast to the statement by Slatyer and McIlroy mentioned above.

The paragraph has been re-written and now includes two paragraphs. Our response to the last three comments is included in this text:

The CR has been used at time scales from hourly to yearly (Brutsaert, 2023, p. 147), but the CR, the Penman equation (1), and the Priestley-Taylor equation (5) most typically use daily- to monthly-average values (McMahon, 2013). It is true that use of time-averages of variables as inputs to non-linear equations can lead to "significant errors" (Slatyer and McIlroy, 1961, p. 3-58). However, CR and the Priestley-Taylor wet-surface equation both assume that that the land-surface conditions and the temperature and humidity in the lower atmosphere are well-adjusted to each other (Brutsaert, 2023, p. 147). The diurnal cycle makes this adjustment unlikely over periods less than 24 hours (McMahon, 2013). Therefore, the approach here is to use daily (monthly) average input values to produce daily (monthly) energy fluxes (e.g., Penman, 1948; McMahon, 2013; Brutsaert, 2023). That is, daily to monthly time scales are suited to these equations, as spatial scales corresponding to small watersheds are suited to saturation-excess runoff (e.g., Chow et al., 1988).

With this dataset, the monthly (daily) mean of reference latent heat flux is 61 W m$^{-2}$ (62 W m$^{-2}$), the median is 58 W m$^{-2}$ (56 W m$^{-2}$) and the standard deviation is 41 W m$^{-2}$ (48 W m$^{-2}$). Thus, the central tendencies for monthly and daily values are similar, but the daily has more spread about the mean.

L587: Please also create a release of the github repo and publish it on a suitable archiving platform (e.g. zenodo.org), to create a static version with its own DOI. See https://www.hydrology-and-earth-system-sciences.net/policies/data_policy.html

We put the data files and the Python code in a zenodo.org repository. Note that we used to have directions on how to run the code directly in GitHub. With zenodo.com those very specific directions are no longer needed, but the "description" on zenodo introduces the project and provides brief instructions.

Report 2:

The manucript was substantially improved and my comments have been responded well. I think it is ready for publishing.
The symbol 'T' are used in both Line 284 and Line 209. Please revise it.

We addressed this near line 209:

A straight line with slope d$e$/d$T_g$=-$\gamma$ (where $T_g$ is a generic temperature variable) represents an isenthalp (line of constant available energy) through ($T_a$, $e_a$) on a graph of temperature ($x$-axis) versus vapor pressure ($e$ on the y-axis).

Report 1:

The authors have addressed the comments. With the mentioned corrections and clarifications in the reply letter and the improvement of figures, the paper is recommended for acceptance.

One additional change was made to the text to clarify the data filtering process. The change was at the start of the third paragraph (in this latest revision) of section 3.3. The new wording is:

> "Months (or days) with eddy covariance values of $H$ and $LE$ less than zero or $(R_n\text{-}G)<0$ were screened out of the dataset; this eliminates periods of strong dry advection that result in negative $H_{ref}$."

---

## Author Response (AR3)

Dear Stan,

Thanks for making sure the project is accessible. I made the following further changes:

1. The data availability statement now reads:

**Data Availability**

All data were downloaded from the FLUXNET website ([http://fluxnet.org](http://fluxnet.org)). Data pre-processing by FLUXNET staff follows procedures described by Pastorello et al. (2020). The datafiles and Python code (Wet_surface_evap_public.ipynb) have been uploaded to Zenodo with doi:10.5281/zenodo.8172604, URL: https://doi.org/10.5281/zenodo.8172604. Instructions are given in the description at Zenodo.

The project is also available from GitHub at [https://github.com/r-crago/FLUXNET_Wet_sfc_evap](https://github.com/r-crago/FLUXNET_Wet_sfc_evap).

2. I changed the data point symbols in Figure 3 so that the different data points can be distinguished by marker shape as well as by color.

Thank you for guiding this manuscript through the process!

Rich (for Joe and Russ)